# Molecular characterization of the conoid complex in *Toxoplasma* reveals its conservation in all apicomplexans, including *Plasmodium* species

Ludek Koreny[1], Mohammad Zeeshan[2], Konstantin Barylyuk[1], Eelco C. Tromer[1], Jolien J. E. van Hooff[3], Declan Brady[2], Huiling Ke[1], Sara Chelaghma[1], David J. P. Ferguson[4,5], Laura Eme[3], Rita Tewari[2]*, Ross F. Waller[1]*

1 Department of Biochemistry, University of Cambridge, Cambridge, United Kingdom, 2 School of Life Sciences, Queens Medical Centre, University of Nottingham, Nottingham, United Kingdom, 3 Université Paris-Saclay, CNRS, AgroParisTech, Ecologie Systématique Evolution, Orsay, France, 4 Nuffield Department of Clinical Laboratory Science, University of Oxford, John Radcliffe Hospital, Oxford, United Kingdom, 5 Department of Biological and Medical Sciences, Faculty of Health and Life Science, Oxford Brookes University, Oxford, United Kingdom

* Rita.Tewari@nottingham.ac.uk (RT); rfw26@cam.ac.uk (RFW)

**Data Availability Statement:** The mass-spectrometry-based proteomics data have been deposited to the ProteomeXchange Consortium via

## Abstract

The apical complex is the instrument of invasion used by apicomplexan parasites, and the conoid is a conspicuous feature of this apparatus found throughout this phylum. The conoid, however, is believed to be heavily reduced or missing from *Plasmodium* species and other members of the class Aconoidasida. Relatively few conoid proteins have previously been identified, making it difficult to address how conserved this feature is throughout the phylum, and whether it is genuinely missing from some major groups. Moreover, parasites such as *Plasmodium* species cycle through 3 invasive forms, and there is the possibility of differential presence of the conoid between these stages. We have applied spatial proteomics and high-resolution microscopy to develop a more complete molecular inventory and understanding of the organisation of conoid-associated proteins in the model apicomplexan *Toxoplasma gondii*. These data revealed molecular conservation of all conoid substructures throughout Apicomplexa, including *Plasmodium*, and even in allied Myzozoa such as *Chromera* and dinoflagellates. We reporter-tagged and observed the expression and location of several conoid complex proteins in the malaria model *P. berghei* and revealed equivalent structures in all of its zoite forms, as well as evidence of molecular differentiation between blood-stage merozoites and the ookinetes and sporozoites of the mosquito vector. Collectively, we show that the conoid is a conserved apicomplexan element at the heart of the invasion mechanisms of these highly successful and often devastating parasites.

the PRIDE partner repository with the dataset identifiers: PXD015269 and https://doi.org/10.6019/PXD015269; and PXD022785 and https://doi.org/10.6019/PXD022785.

**Funding:** This work was supported by UKRI Medical Research Council (MRC) MR/M011690/1 (to LK, RFW); MRC G0900278 (to MZ, DB, RT) and MRC MR/K011782/1 (to MZ, DB, RT); Wellcome Trust 214298/Z/18/Z (to LK, KB, RFW); Wellcome Trust 108441/Z/15/Z (to RFW); Leverhulme Trust ECF-2015-562 (to KB); Isaac Newton Trust ECF-2015-562 (to KB); Herchel Smith (to ECT); UKRI Biotechnology and Biological Sciences Research Council (BBSRC) BB/N017609/1 (to RT, MZ); European Research Council (ERC) 803151 (to JvH, LE). The funders had no role in study design, data collection and analysis, decision to publish, or preparation of the manuscript.

**Competing interests:** The authors have declared that no competing interests exist.

**Abbreviations:** 3D-SIM, 3D structured illumination microscopy; APR, apical polar ring; BioID, proximity-dependent biotin identification; GFP, green fluorescent protein; HMM, Hidden Markov Model; hyperLOPIT, hyperplexed Localisation of Organelle Proteins by Isotope Tagging; IMC, inner membrane complex; MORN, Membrane Occupation and Recognition Nexus; MTOC, microtubule organising centre; SAR, Stramenopila–Alveolata–Rhizaria; SAS6L, SAS6-like; TEM, transmission electron micrograph; U-ExM, ultrastructure expansion microscopy.

## Introduction

It is difficult to imagine a more insidious intrusion upon an organism's integrity than the penetration and occupation of intracellular spaces by another foreign organism. Apicomplexan parasites are masters of this transgression through actively seeking, binding to, and invading the cellular milieu of suitable animal hosts. From here, they manipulate and exploit these cells to promote their growth and onward transmission to other cells and other hosts. The impacts of these infection cycles include major human and animal diseases, such as malaria, toxoplasmosis and cryptosporidiosis in humans, and a spectrum of other diseases in livestock and wild animals [1–4]. The course of both human history and evolution has been shaped by these ubiquitous specialist parasites.

Key to the successful parasitic strategies of apicomplexans is the apical complex—a specialisation of the cell apical cortex that coordinates the interaction and penetration of host cells [5]. Most of the apicomplexan cell is encased in a pellicle structure of flattened membrane vesicles beneath the plasma membrane, as are all members of the infrakingdom Alveolata including dinoflagellates and ciliates [6]. These "alveoli" sacs are supported by robust proteinaceous networks, and collectively, this inner membrane complex (or IMC, as it is called in apicomplexans) provides shape and protection to the cell, as well as other functions such as gliding motility in apicomplexans by IMC-anchored motors [7]. The IMC, however, is a general obstruction to other cellular functions that occur at the plasma membrane, including exocytosis and endocytosis [8]. Thus, the apical complex has evolved alongside the IMC to provide a location for these functions. When apicomplexans attack their host's cells, the apical complex is the site of exocytosis; first of host-recognition and host-binding molecules, and subsequently of molecules injected into the host that create a platform in its plasma membrane for parasite penetration [9,10]. In infections such as those that humans suffer from, upon host cell invasion, further exocytosed molecules create a modified environment in the host cell that facilitate the parasite's growth, reproduction, and protection from the host's immune system. In many gregarine apicomplexans, on the other hand, only partial penetration of the host occurs and the parasite endocytoses host cytosol via their embedded apical complex [11]. Near relatives of Apicomplexa, such as colpodellids and some dinoflagellates, similarly feed on prey and host cells through their apical complex—such is the apparent antiquity of this cell feature [12,13]. The apical complex is thus a coordination of the cell cytoskeleton that defines an available disc of the plasma membrane that is unobscured by the IMC for vesicular trafficking machinery to deliver and exchange with the extracellular environment. A protuberance of the cell at the apical complex also provides mechanical properties to this important site.

Functional studies of the apical complex have occurred in select experimentally amenable taxa, mostly *Toxoplasma* and *Plasmodium*, but a mechanistic understanding of this cell feature or its conservation is still in its infancy. Rather, the apical complex is most broadly understood from ultrastructural studies that show apical rings as the basis of this structure. An apical polar ring (APR1) coordinates the apical margin of the IMC, and a second APR (APR2) acts as a microtubule organising centre (MTOC) for the subpellicular microtubules (Fig 1) [14–16]. Within this opening created by the APRs are further rings, a notable one being the "conoid" that is conspicuous throughout much of Apicomplexa [17]. The conoid is a tapered tubular structure of variable length and cone pitch. It interacts intimately with secretory organelles including micronemes, rhoptries, and other vesicles that penetrate and occupy its lumen [18,19]. An open conoid (often referred to as "pseudoconoid") seen in *Perkinsus*, another parasite and close relative of Apicomplexa, even has microneme-like vesicles physically tethered to it, and in Coccidia a pair of intraconoidal microtubules is lined by a chain of microvesicles [18,20]. In gregarines, endocytosis occurs through the conoid aperture [11]. Thus, while the

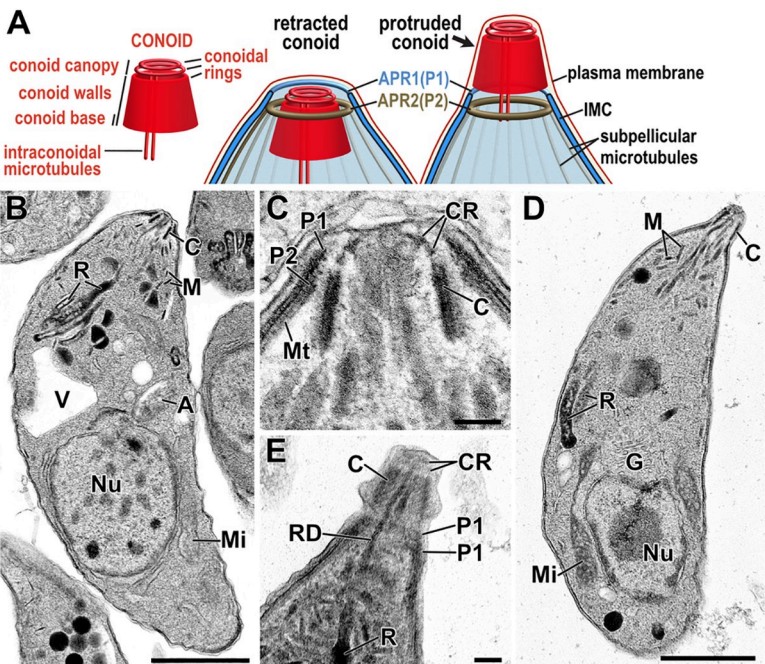

**Fig 1. Conoid complex features in *Toxoplasma* tachyzoites.** (A) Schematic of the recognised components of the conoid and their location within the apical structures of the cell pellicle in either retracted or protruded states. (B–E) Transmission electron micrographs of *T. gondii* tachyzoites with conoid either retracted (B, C) or protruded (D, E). Tubulin filaments of the conoid walls are evident in tangential section (E) and 2 CRs of the conoid canopy are evident at the conoid's anterior end (D, E). The conoid is surrounded by 2 APRs (P1 and P2) formed by the anterior aspect of the IMC and anchoring the subpellicular microtubules (Mt), respectively. RDs can be seen running to the apex through conoid (E). Scale bars represent 1 μm (B, D) and 100 nm (C, E). A, apicoplast; APR, apical polar ring; C, conoid; CR, conoidal ring; G, Golgi apparatus; IMC, inner membrane complex; M, micronemes; Mi, mitochondrion; Mt, microtubule; Nu, nucleus; R, rhoptries; RD, rhoptry duct; V, plant-like vacuole.

APRs appear to play chiefly structural organising roles, the conoid is closely associated with the events and routes of vesicular trafficking—delivery and in some cases uptake.

In most known conoids, the walls of the conoid have a spiralling fibrous presentation by electron microscopy (Fig 1E), a trait that is chiefly attributed to the presence of tubulin polymers [16,17,21]. In the *Toxoplasma* conoid, tubulin forms unusual open fibres with a comma-shaped profile [21]. The ancestral state of conoid tubulin, however, is likely canonical microtubules as seen in gregarines, *Chromera*, and other apicomplexan relatives [11,13,22]. It is unclear if the modified tubulin fibres of the *Toxoplasma* conoid arose specifically within coccidians or are more widespread in apicomplexans due to the limits of resolution or preservation of this dense structural feature. Nevertheless, this tubulin component demonstrates a degree of plasticity of the conoid structure. Electron microscopy shows that the tubulin fibres are embedded in electron-dense material, evidence of further conoid proteins (Fig 1C) [14,17,23]. This matrix extends to an open apical cover described as a "delicate osmophilic... canopy" by Scholtzseck and colleagues (1970) within which 2 conoidal rings are often seen (Fig 1A, 1C and 1E). These rings are now frequently referred to as "preconoidal rings;" however, in recognition of the continuity of conoid ultrastructure from spiral reinforced walls to canopy rings, this entire structure was designated as the conoid and the rings as "conoidal rings" [17]. The apical conoid canopy is in closest contact, and probably interacts, with the cell plasma membrane [14,23]. Electron microscopy does not reveal any direct attachment fibres or structures from the conoid to the plasma membrane at its apex, or to the IMC at its base. However, in

*Toxoplasma*, it is known that at least one protein (RNG2) links the conoid directly to the APR2 [24]; thus, there is evidence of molecular architecture beyond that observed by ultrastructure.

A predicted structural deviation to the apical complex in Apicomplexa is the interpretation of loss of the conoid in some groups, a state enshrined within the class name Aconoidasida. This class contains 2 important groups: Haemosporida, such as *Plasmodium* spp., and Piroplasmida. Aconoidasida are considered to have either completely lost the conoid (e.g., *Babesia*, *Theileria*), or at least lost it from multiple zoite stages, e.g., *Plasmodium* spp. stages other than the ookinete. However, while conoids have been attributed to micrographs of ookinete stages in some *Plasmodium* spp., in other studies, these are alternatively labelled as "apical polar rings" [17,25–27], and the prevailing understanding of many is that a conoid was lost outright.

The uncertainty over whether the conoid is present in Aconoidasida is a product of 2 problems. One is that we have little insight into the function of the conoid, so the consequences of its loss are difficult to predict. The other is that we still know relatively little of the molecular composition of the conoid that would allow the objective testing for the presence of a homologous structure [5]. The conspicuous ultrastructure of conoids such as those of coccidians draw attention to tubulin being a major element; however, it is known that there are other conoid proteins responsible for its assembly, stability, and function during invasion [24,28–35]. To test if a homologous conoid cell feature is present in Aconoidasida, but cryptic by traditional microscopy techniques, fuller knowledge of the molecules that characterise this feature in a "classic" conoid model is needed. In our study, we have sought such knowledge for the *Toxoplasma gondii* conoid using multiple proteomic approaches. We then asked if these conoid-associated proteins are present in similar locations within Aconoidasida using the model *Plasmodium berghei* to investigate each of its zoite forms: ookinetes, sporozoites, and merozoites. In doing so, we address the question of what common machinery underpins the mechanisms of invasion and host exploitation that are central to these parasites' lifestyles and impact. Our data also explore the antiquity of this machinery and its presence in relatives outside of Apicomplexa.

## Results

### Spatial proteomic methods identify new candidate conoid proteins

To expand our knowledge of the proteins that contribute to conoid structure and function, we applied multiple spatial proteomics discovery approaches. The primary method used was hyperplexed Localisation of Organelle Proteins by Isotope Tagging (hyperLOPIT) that we have recently applied to extracellular *T. gondii* tachyzoites [36,37]. This approach entailed generating hyperLOPIT datasets from 3 independent samples. In each sample, mechanically disrupted cells were dispersed on density gradients, and the distinct abundance distribution profiles formed by different subcellular structures and compartments were used to identify proteins that belong to common cellular niches. Combining the data from the 3 experiments provided enhanced discriminating power of protein location assignments, and from the 3,832 proteins that were measured in all 3 samples we reported 63 proteins assigned to one of the 2 apical protein clusters, *apical 1* and *apical 2*, above a 99% probability threshold. Another 13 proteins were assigned to these clusters but below this high-confidence cut-off. The 2 high-confidence clusters were verified as comprising proteins specific to the structures associated with the conoid, APRs, and "apical cap" of the IMC [36]. In addition to the 3,832 proteins that we reported in this high-resolution spatial proteome [36], a further 1,013 proteins were quantified in either only two or one of the hyperLOPIT datasets due to the stochasticity of mass spectrometry sampling. While assignment of these proteins consequently had less data support, a further 32 proteins were assigned to the *apical* clusters (not reported in the Barylyuk's and

colleagues' study) from analysis of either the pairs of LOPIT experiments or the single experiments. From these analyses, 95 proteins were assigned as putative apical proteins across these hyperLOPIT samples (S1 Table).

Of the 95 putative apical protein assignments by hyperLOPIT, 13 had been validated as being located to the very apex of the cell during our hyperLOPIT study [36], 23 with this same specific apical location by us or others previously, and 21 proteins were known to be specific to the apical cap or other IMC elements (S1 Table and refs therein). This left a further 38 new protein candidates for which there was no independent validation of their apical location. To bolster support for conoid-specific location, we applied a second spatial proteomic strategy, proximity-dependent biotinylating and pulldown (BioID) [38]. We made 3 apical BioID "baits" by endogenous 3′ gene fusions with coding sequence for the promiscuous biotin-ligase BirA\*. Two baits were known conoid markers: SAS6-like (SAS6L), a protein previously attributed to the conoid canopy ("preconoidal rings") in *T. gondii* [39], but, by super-resolution imaging, we observe this protein located in the body of the conoid (see below); and RNG2 where the C-terminus of this large protein is anchored to the APR2 that is in close proximity to the apical end of the conoid in intracellular parasites [24]. A third bait protein is an otherwise uncharacterised Membrane Occupation and Recognition Nexus (MORN) domain-containing protein (TGME49_245710) that we call MORN3. In an independent study of MORN proteins, we identified MORN3's location as being throughout the IMC but with greatest abundance in a band at the apical cap, although excluded from the very apex where the conoid is located (Fig 2A). Using these 3 BioID baits, we rationalised that SAS6L and RNG2 proximal proteins would be enriched for those near the conoid, and MORN3 proximal proteins would be enriched for apical cap and IMC proteins but not for conoid-proximal proteins.

*T. gondii* cell lines expressing each of the BioID bait proteins were grown for 24 hours in host cells with elevated exogenous biotin. Streptavidin-detection of biotinylated proteins on western blots showed unique banding patterns of each cell line and when compared to parental controls (cells lacking BirA fusions) (Fig 2B). Biotinylated proteins from each cell line were then purified on a streptavidin matrix and analysed by mass spectrometry. Proteins enriched ≥3-fold compared to the control, or detected in the bait cell lines but not in the control, are indicated in S1 Table. Of the hyperLOPIT-assigned *apical* proteins, 25 were also detected by BioID with both SAS6L and RNG2 but not MORN3, and these included known conoid-associated proteins (e.g., MyoH, CPH1, CIP2, CIP3, SAS6L, RNG2). Seven proteins were BioID-detected by MORN3 but not SAS6L or RNG2, and these are all known apical cap or IMC proteins (AC4, AC8, AC9, AC10, AAP5, IMC9, IMC11). These data indicate that the BioID spatial proteomics indeed enrich for apical proteins, with the differences between SAS6L/RNG2 and MORN3 labelling providing a level of discrimination for conoid-associated proteins when compared to apical cap proteins.

## Validation of conoid proteins and determination of their substructural location

We confirmed the identification of new apical complex proteins in the region of the conoid in the hyperLOPIT study by endogenous 3′-tagging of candidate genes with reporters [36]. Imaging by wide-field fluorescence microscopy showed 13 of these proteins confined to a single small punctum at the extreme apex of the cell (Table 1). To test if our expanded hyperLOPIT analysis, including proteins with less hyperLOPIT *apical* assignment support, contained further proteins at the apical tip, seven of these were tagged by the same method (S1 Table: TGME49_274160, TGME49_219070, TGME49_209200, TGME49_274120, TGME49_250840, TGME49_219500, TGME49_284620). All of these proteins were observed to show the same

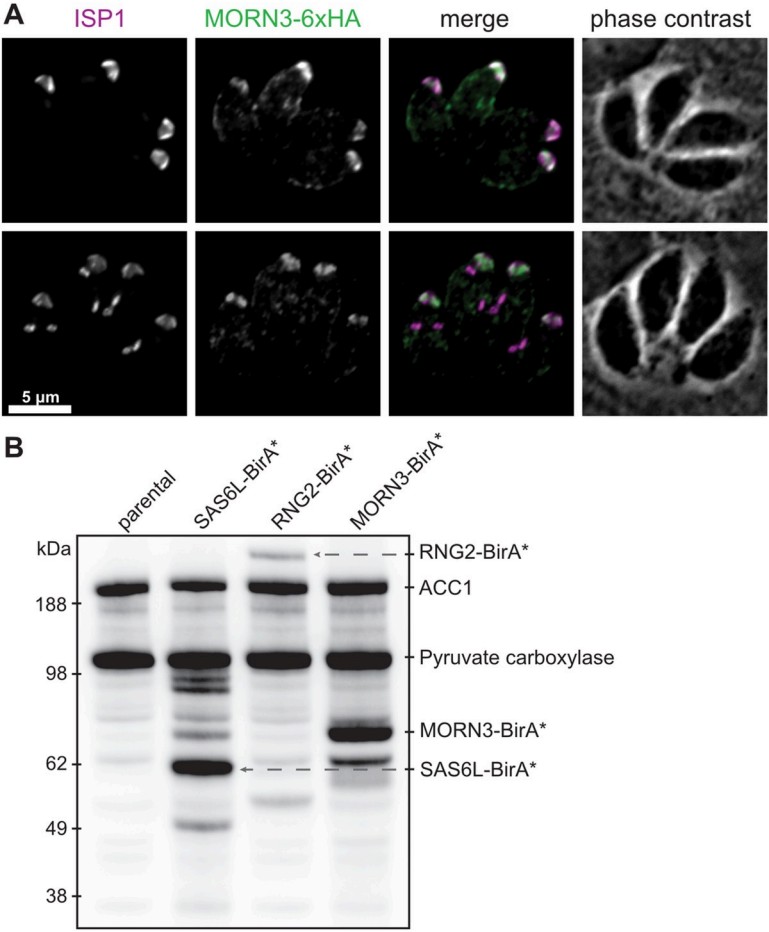

**Fig 2. Apically targeted BioID using baits RNG2, SAS6L, and new apical cap protein MORN3.** (A)
Immunodetection of HA-tagged MORN3 in *T. gondii* intracellular parasites costained for apical cap-marker ISP1.
Upper panels show mature cells; lower panels show internal daughter pellicle buds forming during the early stages of
endodyogeny. Scale bar = 5 μm. (B) Steptavidin-detection of biotinylated proteins after 24 hours of growth with
elevated biotin. Native biotinylated proteins ACC1 and pyruvate carboxylase are seen in the parental control (lacking
BirA*) and in the BioID bait cell lines. Additional biotinylated proteins are seen in each of the bait cell lines grown
with elevated biotin, including self-biotinylation of the bait fusion. ACC1, Acetyl-CoA carboxylase; BioID, proximity-
dependent biotin identification; HA, hemagglutinin; MORN3, Membrane Occupation and Recognition Nexus 3;
SAS6L, SAS6-like.

extreme apical location (S1, S3, and S4 Figs). All of these proteins that were independently
tested for location were previously uncharacterised and were selected only based on sharing
orthologues with other apicomplexan clades (see below), strong phenotypes identified by a
genome-wide knockout screen [40] or presence of conserved domains that might ultimately
provide clues of molecular function. Among the hyperLOPIT *apical*-assigned proteins tagged
and located in either this or the Barylyuk's and colleagues' (2020) study all located to the apical
structures of the cell.

The conoid of *T. gondii* is a small structure (approximately 400 × 500 nm) in close proxim-
ity to the APRs (Fig 1) so widefield fluorescence microscopy is less able to distinguish between
proteins at either of these specific structures, nor subdomains of the conoid itself. To deter-
mine the specific locations of our conoid-proximal proteins, we employed 3D structured illu-
mination microscopy (3D-SIM) super-resolution imaging in cells coexpressing either SAS6L
or RNG2 with C-terminal epitope reporters. By 3D-SIM, we observe SAS6L with C-terminal

**Table 1.** *Toxoplasma* conoid-associated proteins and *Plasmodium* orthologues.

| [a] T. gondii ME49 | Protein Name | [b] Known Localization | * Ref for Localization | hyperLOPIT | [c] Proteomics SAS6-like | BioID RNG2 | MORN3 | [a] P. berghei ANKA | [d] zoite stage Ookinete | Sporozoite | Merozoite | [e] Mutant Fitness Scores T. gondii | P. falciparum | [f] Conserved Domains |
|---|---|---|---|---|---|---|---|---|---|---|---|---|---|---|
| 219070 | | CCP | TS | + | | | | 1025300 | • | • | • | −2.20 | −1.63 | EF, Crp, CAP_ED |
| 274160 | | CCP | TS | + | | | | 1313300 | • | • | • | −2.80 | −3.14 | |
| 209200 | | CCP | TS | + | | | | 1436500 | | | | −1.55 | −2.53 | EF |
| 284620 | | CCR | TS | + | | | | 1316900 | | | | −1.02 | −1.83 | LRR |
| 253600 | | CCR | [36],TS | + | | | | 0713200 | | | | −2.40 | −2.56 | |
| 306350 | | CCR | [36],TS | + | | | | 1347000 | • | • | • | −0.84 | −0.98 | PH-like |
| 202120 | ICAP16 | CCR | [40],TS | | | • | | 1419000 | • | • | • | −2.10 | −3.04 | |
| 250340 | Centrin 2 | CCR+AA | [44,45,46] | + | | | | 1310400 | | | | −4.41 | −2.08 | EF |
| 222350 | | conoid body | [31,36],TS | + | • | • | | 1229900 | | | | −1.31 | 0.02 | |
| 274120 | | conoid body | [31],TS | + | • | • | | 0310700 | • | • | ○ | 0.64 | −0.38 | |
| 291880 | | conoid body | [36],TS | + | • | • | | 0616200 | | | | 1.77 | 0.15 | |
| 297180 | | conoid body | [36],TS | + | • | • | | | | | | −1.52 | | CRAL-TRIO |
| 301420 | SAS6L | conoid body | [39,42],TS | + | • | • | | 1414900 | • | • | ○ | −1.62 | −0.80 | SAS6_N |
| 243250 | MyoH | conoid body | [28] | + | • | • | | | | | | −3.94 | | Myo, Cal, RCC1 |
| 295450 | DIP13 | conoid body | [43] | | | | | 1141900 | | | | 0.67 | | |
| 256030 | DCX | conoid body | [32,33] | + | • | • | • | 1232600 | • | • | | −5.03 | −2.46 | UBQ, p25-α |
| 226040 | CAM3 | conoid body | [30] | + | | | | | | | | −3.25 | | EF |
| 262010 | CAM2 | conoid body | [30,33] | + | | | | | | | | −0.81 | | EF |
| 246930 | CAM1 | conoid body | [28,44] | + | | | | | | | | 1.09 | | EF |
| 246720 | | conoid base | [31,36],TS | + | • | • | | 0109800 | • | • | • | 0.24 | −2.83 | EF |
| 258090 | | conoid base | [31,36],TS | + | • | • | | 1216300 | • | • | ○ | −1.34 | −0.40 | |
| 266630 | CPH1 | conoid base | [31,36],TS | + | • | • | | 0620600 | | | | −4.16 | −0.42 | ANK |
| 244470 | RNG2 | conoid base + APR2 | [24,34],TS | + | • | • | | | | | | −4.21 | | |
| 239300 | ICMAP1 | ICMT | [47] | + | | • | | | • | • | | −0.74 | | |
| 208340 | | APR1 | [36],TS | + | • | • | | 907700 | • | • | • | −0.81 | −3.05 | PH-like |
| 250840 | MLC3 | APR1 | [28],TS | + | • | • | | | | | | −1.91 | | EF |
| 219500 | | APR1/2 | TS | + | • | • | | 0919400 | | | | 0.00 | −2.89 | Ribonucl-like |
| 227000 | | APR1/2 | [31,36],TS | + | • | • | | 0510100 | | | | −3.17 | −0.24 | |
| 267370 | Kinesin A | APR1/2 | [29] | + | • | | | | | | | −2.70 | | Kinesin |
| 278780 | | APR2 | [36],TS | + | • | • | • | | | | | −2.77 | | UBQ |
| 320030 | | APR2 | [31,36],TS | + | • | • | • | 1334800 | • | • | ○ | −0.19 | −2.43 | |
| 243545 | RNG1 | APR2 | [34,48] | | • | | | | | | | 2.54 | | |
| 315510 | APR1 | APR2 | [29] | + | • | • | | | | | | −0.05 | | |
| 292120 | MORN2 | apical PM ring | [36],TS | | | | | | | | | −0.04 | | MORN |
| 226990 | | apex | [31] | + | • | | • | | | | | 1.41 | | |

(*Continued*)

**Table 1.** (Continued)

| [a]T. gondii ME49 | Protein Name | [b]Known Localization | *Ref for Localization | [c]Proteomics hyperLOPIT | BioID SAS6-like | BioID RNG2 | BioID MORN3 | [a]P. berghei ANKA | [d]zoite stage Ookinete | Sporozoite | Merozoite | [e]Mutant Fitness Scores T. gondii | P. falciparum | [f]Conserved Domains |
|---|---|---|---|---|---|---|---|---|---|---|---|---|---|---|
| 234270 | | apex | [31] | + | • | | | | | | | −0.44 | | |
| 254870 | | apex | [31] | + | | | | | | | | 0.64 | | TerD |
| 255895 | | apex | [31] | + | • | • | | | | | | 0.23 | | |
| **295420** | | apex | [31,36] | + | • | • | • | | | | | −1.57 | | TerD |
| 313780 | | apex | [31] | + | • | • | • | | | | | 0.71 | | |
| 291020 | MyoL | apex | [51] | | | | | 1435500 | | | | −1.83 | −2.97 | Myo, RCC1 |
| 239560 | MyoE | apex | [51] | + | | | | | | | | 0.11 | | Myo |
| 315780 | MLC7 | apex | [28] | + | | | | 0514800 | | | | −0.12 | −3.04 | EF |
| 311260 | MLC5 | apex | [28] | + | | | | | | | | −0.33 | | EF |
| 209890 | ICAP4 | apex | [40] | + | | | | 1439000 | | | | −4.84 | −2.17 | |
| 312630 | GAC | apex | [53] | | | | | 1137800 | • | • | | −3.53 | −3.05 | ARM |
| 206430 | FRM1 | apex | [49] | + | | | | 1245300 | | | • | −3.24 | −2.92 | TPR |
| 252880 | CRMP | apex | [54] | + | • | • | | | | | | −2.35 | | Ax_dyn_light |
| 225020 | CIP3 | apex | [31] | + | • | • | | 1309800 | | | | −2.78 | −1.04 | |
| 257300 | CIP2 | apex | [31] | + | • | • | | | | | | −2.49 | | |
| 234250 | CIP1 | apex | [31] | | • | • | | 1423000 | | | | −2.02 | −2.77 | |
| 210810 | CAP1 | apex | [55] | | | | | | | | | −0.73 | | |
| 216080 | AKMT | apex | [50] | | • | • | | 0932500 | | | | −4.30 | −2.08 | SET |
| 310070 | AAMT | apex | [52] | | | | | 1318900 | | | | −1.22 | −2.05 | Methyltrans |

[a] Proteins with location data by microscopy in this and the Barylyuk's and colleagues' (2020) study shown in **bold**.

[b] Known localization defined as "apex" when low-resolution imaging only has identified a punctum at the apex of the cell. AA, apical annuli; APR, apical polar ring; APR1/2 indicates intermediate position between the two rings; CCP, conoid canopy punctum; CCR, conoid canopy ring; ICMT, intraconoidal microtubules; PM, plasma membrane.

[c] Proteomic data: +, proteins represented in the hyperLOPIT data; •, BioID–detection of a protein with a given "bait."

[d] P. berghei zoite stage presence (•) or absence (o) of detectible protein–GFP expression by live-cell imaging.

[e] Mutant phenotype fitness scores where more strongly negative scores indicate increasingly detrimental competitive growth in in vitro culture conditions for *T. gondii* tachyzoites [40] or *P. falciparum* blood-stage parasites [41].

[f] Conserved domain abbreviations: ARM, armadillo repeat; Ax_dyn_light, Axonemal dynein light chain; Cal, Calmodulin-binding motif; EF, EF-hand; LLR, leucine-rich repeat; Methyltrans, Methyltransferase; Myo, myosin; TPR, Tetratricopeptide repeat; Ribonucl-like, Ribonuclease-like.

* References for localization data: TS, this study.

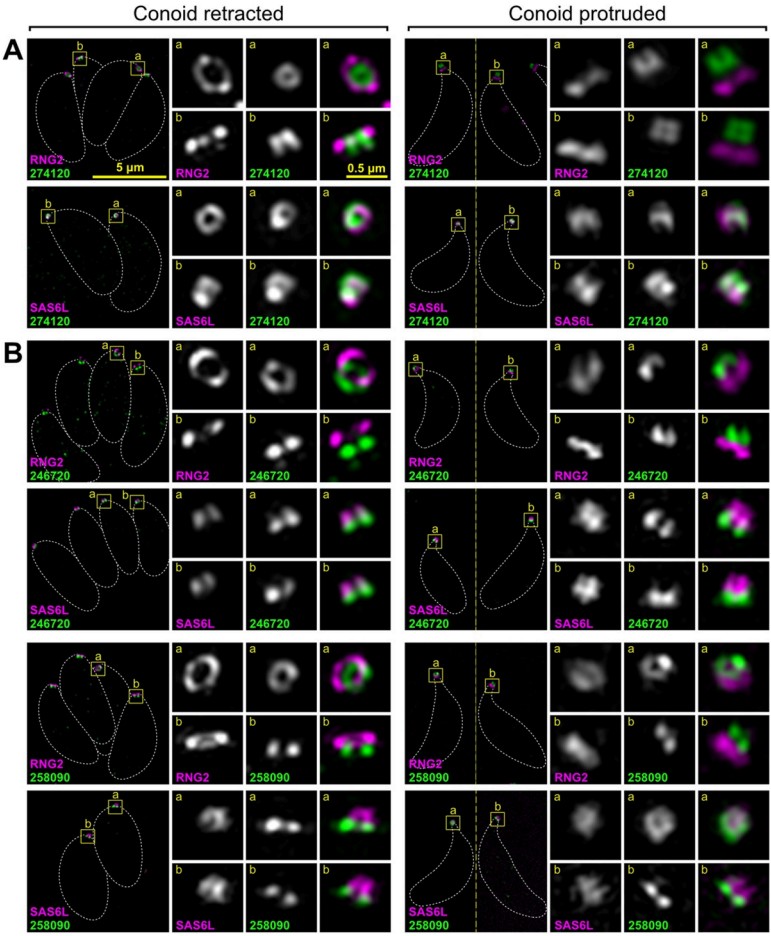

**Fig 3. Super-resolution imaging of *T. gondii* proteins at the conoid body and base.** Immunodetection of HA-tagged conoid proteins (green) in cells coexpressing either APR marker RNG2 or conoid marker SAS6L (magenta) imaged either with conoids retracted with parasites within the host cell, or with conoids protruded in extracellular parasites. (A) Example of protein specific to the conoid body and (B) examples of proteins specific to the conoid base. See S2 and S3 Figs for further examples. All panels are at the same scale, scale bar = 5 μm, with zoomed inset from white boxed regions (inset scale bar = 0.5 μm). Dashed white lines indicate the cell boundary. APR, apical polar ring; HA, hemagglutinin; SAS6L, SAS6-like.

V5 epitope tag to locate to the tapered walls of the conoid (Figs 3–5), rather than exclusively to the rings of the apical conoid canopy as was previously reported for YFP-tagged SAS6L [39]. The fluorescence imaging used in the de Leon's and colleagues' study was limited to lower resolution widefield microscopy. Immuno-TEM was employed also, however, contrary to the authors' conclusions, did show YFP presence throughout transverse and oblique sections of the conoid consistent with our detection of SAS6L throughout the conoid body. RNG2 C-terminal reporters locate within the region of the APRs and, given its adherence to the apical ends of the subpellicular microtubules after detergent-extraction, it was presumed to be an APR2 location [14,24]. These 2 markers, for the mobile conoid body and apex of the IMC, provided spatial definition of the relative positions of the new proteins. Moreover, we exploited the motility of the conoid with respect to the apical polar rings to further discriminate which structures our new proteins were associated with by imaging with the conoid in both retracted and protruded positions (Fig 1).

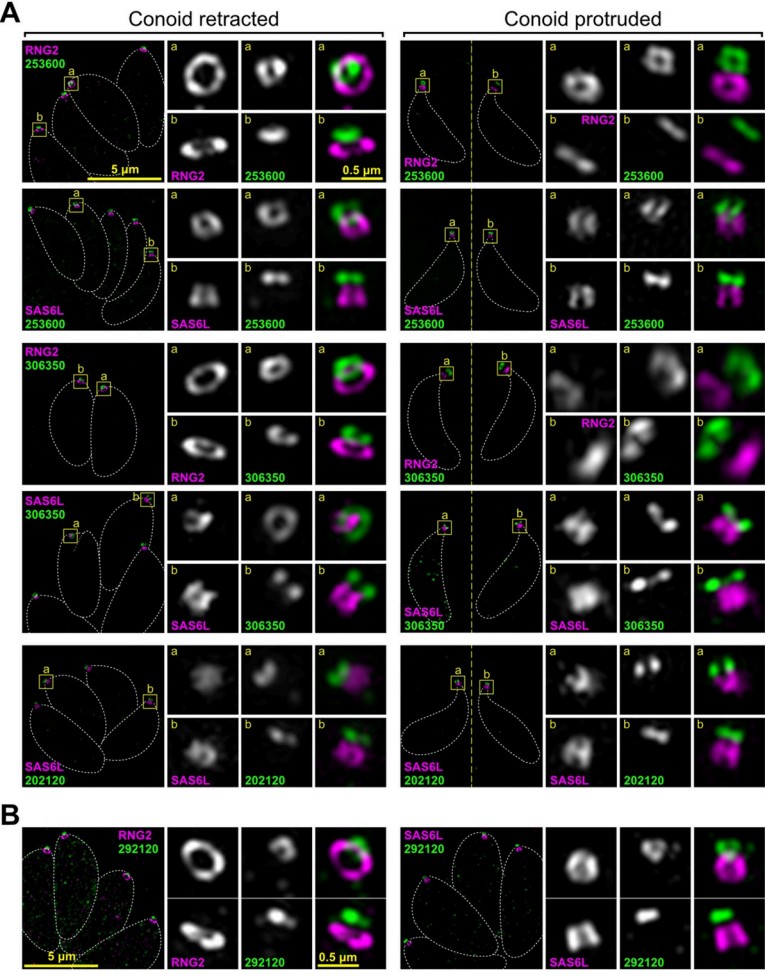

**Fig 4. Super-resolution imaging of *T. gondii* proteins at the conoid canopy rings and MORN2 at the plasma membrane.** (A) Examples of proteins specific to the conoid canopy rings. (B) Peripheral membrane protein (cytosolic leaflet) MORN2 in intracellular parasites. Immunodetection of HA-tagged proteins as for Fig 3. See S3 Fig for further examples. All panels are at the same scale, scale bar = 5 μm, with zoomed inset from white boxed regions (inset scale bar = 0.5 μm). HA, hemagglutinin; MORN2, Membrane Occupation and Recognition Nexus 2.

Using the above strategy, 4 proteins were seen to be specific to the conoid body (TGME49_274120, TGME49_222350, TGME49_297180, TGME49_291880), the last of which was most enriched in the apical half of the conoid body (Fig 3A and S2 Fig). A further 3 proteins were either specific to (TGME49_246720, TGME49_258090) or enriched at (TGME49_266630, or "CPH1") the conoid base (Fig 3B and S3A Fig). Seven proteins were observed to be associated with the conoid canopy, four resolving as small rings (TGME49_253600, TGME49_306350, TGME49_202120, TGME49_284620) (Fig 4A and S3B Fig) and three a punctum too small to resolve (TGME49_274160, TGME49_219070, TGME49_209200) (Fig 5A and S3C Fig). All of these proteins showed motility with SAS6L during conoid protrusion and retraction consistent with being attached to the conoid.

In addition to the conoid proteins, we identified *apical* proteins associated with the APRs. Two proteins collocated with RNG2 at APR2 (TGME49_320030, TGME49_278780), whereas 2 proteins (TGME49_208340, TGME49_250840) were distinctly anterior to RNG2 suggesting they might locate to the APR1 at the extreme apex of the IMC (Fig 5B, S4 Fig). The epitope

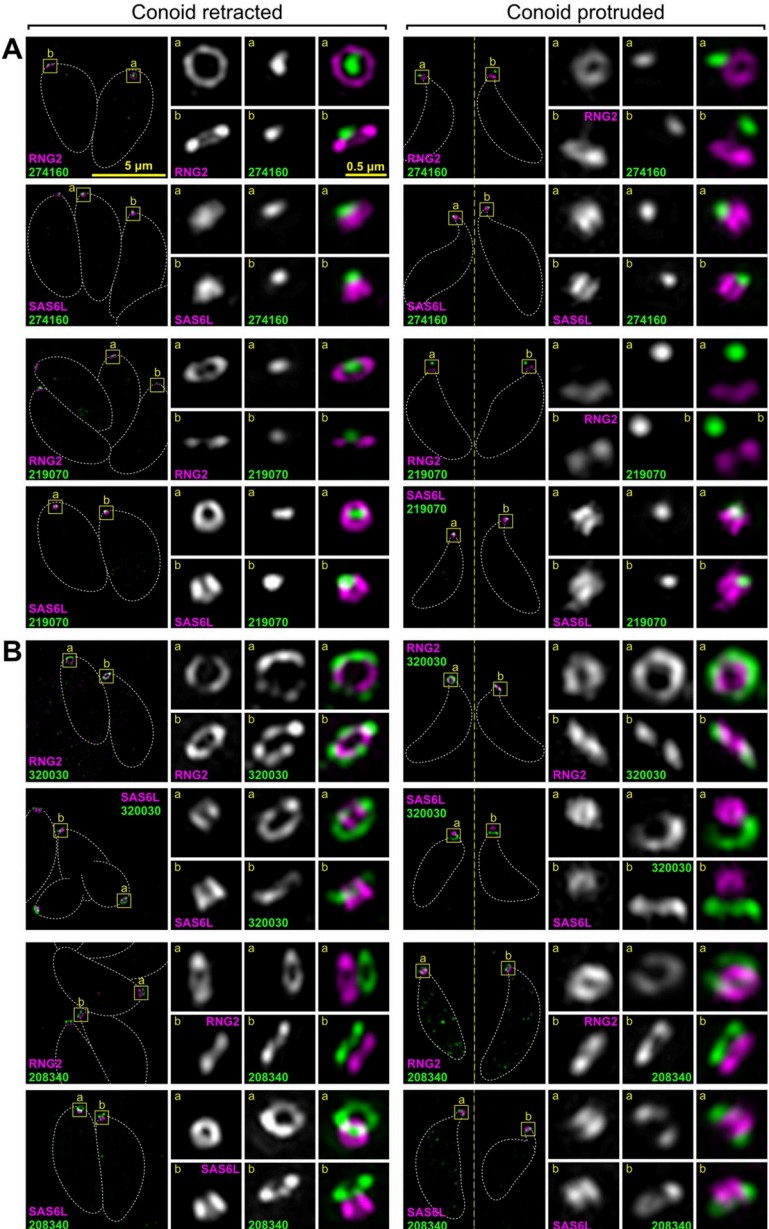

**Fig 5. Super-resolution imaging of *T. gondii* proteins at conoid canopy puncta and the apical polar rings.** Immunodetection of HA-tagged proteins as for Fig 3. (A) Examples of protein specific to the conoid canopy puncta. (B) Examples of proteins specific to the apical polar rings in the vicinity of APR1 (TGME49_208340) and APR2 (TGME49_320030). See S3 and S4 Figs for further examples. All panels are at the same scale, scale bar = 5 µm, with zoomed inset from white boxed regions (inset scale bar = 0.5 µm). APR1, apical polar ring 1; APR2, apical polar ring 2; HA, hemagglutinin.

markers for 2 further proteins (TGME49_227000, TGME49_219500) showed intermediate positions between APR1 and APR2 (S4 Fig). All of these APR proteins were static with respect to RNG2 when the conoid was protruded.

All protein locations determined by super-resolution microscopy were consistent with proximity to, and detection by, the 3 BioID baits. (1) SAS6L/RNG2-positive but MORN3-negative signals detected conoid-proximal proteins: proteins of the conoid body, base, and one

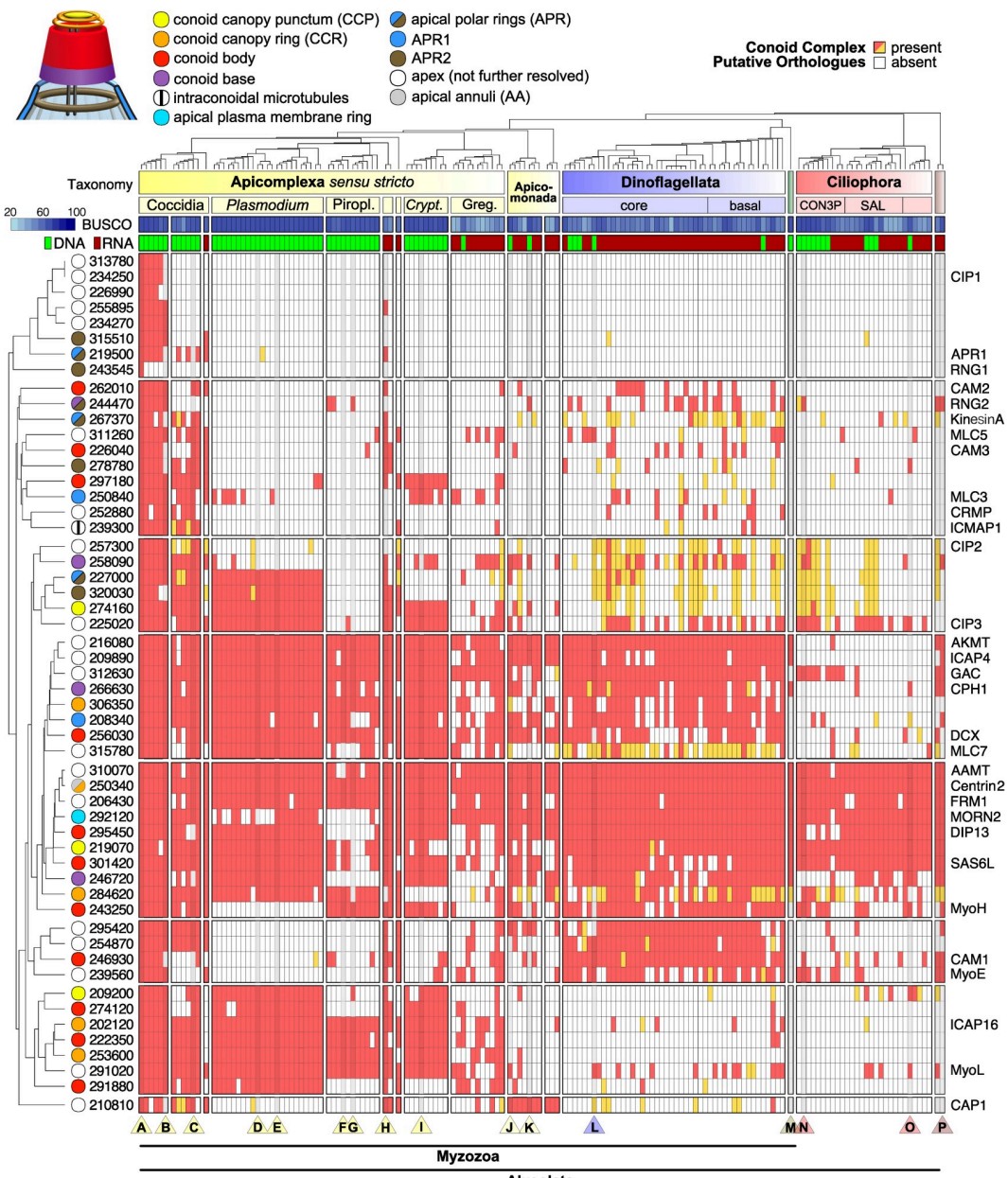

**Fig 6. Heatmap indicating conservation of conoid-associated proteins among Alveolata.** Presence (red, orange) and absence (white) of putative orthologues of 54 *T. gondii* conoid-associated proteins (Table 1) in 157 surveyed Alveolata species (see S5 Fig for taxa, S3 Table for orthologue numbers, and S1 Data for orthologue sequences). ToxoDB protein numbers (left) and existing protein names (right) are shown. In case of a presence, the taxon either contains at least one homologous sequence that has the *T. gondii* protein as its best BLASTp match (red) or it has only homologous sequences that were obtained via sensitive HMMer searches but that did not retrieve a *T. gondii* match by BLASTp (orange), indicative of more divergent homologues (see Methods). The proteins are shown clustered according to their binary (presence-absence) patterns across the Alveolata. Known protein locations in *T. gondii* are indicated by colour (see key) where "apex" indicates low-resolution imaging of an apical punctum only. The species tree (top) shows phylogenetic relationships and major clades: Piropl., Piroplasmida; *Crypt.*, *Cryptosporidium*; Greg., Gregarinasina; green shading, Perkinsozoa; brown shading, Colponemidia. Columns for species of interest are darkened and indicated by a triangle at the bottom of the figure (A–P)–A: *Toxoplasma gondii*; B: *Sarcocystis neurona*; C: *Eimeria tenella*; D: *Plasmodium berghei*; E: *Plasmodium falciparum*; F: *Babesia bovis*; G: *Theileria parva*; H: *Nephromyces* sp.; I: *Cryptosporidium parvum*; J: *Chromera velia*; K: *Vitrella brassicaformis*; L: *Symbiodinium microadriaticum*; M: *Perkinsus marinus*; N: *Tetrahymena thermophila*; O: *Stentor coeruleus*; P: Colponemid sp.. For each species the source of the protein predictions is indicated: genome (DNA, green) or transcriptome (RNA, dark red), along with BUSCO score as estimates of percentage completeness. AA, apical annuli; APR, apical polar ring; CCP, conoid canopy punctum; CCR, conoid canopy ring.

in the canopy (TGME49_202120); proteins of APR1; and the proteins between APR1 and APR2. (2) Proteins negative for all 3 baits occurred at the conoid canopy apparently out of reach of even SAS6L. (3) Proteins detected by all 3 baits were at the APR2 where the 3 proteins converge. Thus, these data suggest that the combination of spatial proteomics methods used provided an effective enrichment of conoid-proximal proteins.

During the hyperLOPIT validation of proteins assigned as *Plasma Membrane–peripheral 2* (on the cytosolic leaflet), one protein, MORN2, was found to be enriched as an apical punctum [36]. Given the close proximity of the conoid apex to the plasma membrane, and unknown molecular interactions between these cell structures that might occur there, we examined the location of MORN2 by 3D-SIM. MORN2 was seen as a small ring anterior to the conoid with a discernible gap between it and the SAS6L-labelled conoid body (Fig 4B). This location is consistent with MORN2 being associated with the plasma membrane and potentially forming a continuum of the annular structures through from the APRs, conoid base, body, and canopy, to the apical plasma membrane.

## Evolutionary conservation of *Toxoplasma* conoid proteins throughout Alveolata

Using the expanded knowledge of conoid-associated proteins determined in this study, and previously identified conoid proteins, we then asked the following questions. How conserved is the *T. gondii* conoid proteome in other apicomplexans and related alveolate lineages (i.e., Apicomonada, Dinoflagellata, Ciliophora)? Is there genomic evidence of conoid presence in Aconoidasida taxa despite the suggestion that this feature was lost from this class of Apicomplexa? To test for the presence or absence of conoid protein orthologues, including highly divergent ones, we used a powerful Hidden Markov Model (HMM) profiling strategy. Briefly, the *T. gondii* apical proteins were first assigned to clusters of predicted orthologues (orthogroups) along with proteins of 419 taxa belonging to the Stramenopila–Alveolata–Rhizaria (SAR) clade using the OrthoFinder algorithm [56]. The sequences of each orthogroup were then used for sensitive detection of divergent homologues in all 419 taxa using HMM profile searches. To exclude putative paralogues and spurious matches from the potential over-sensitivity of the HMM approach from these expanded orthogroups, all collected homologues were used as queries for reverse BLASTp-searches against the *T. gondii* proteome; only homologues that recovered the specific *T. gondii* conoid-associated protein as their best match (Fig 6, red), or no *T. gondii* proteins (potentially indicative of fast-evolving protein families, Fig 6 orange) were retained as putative orthologues.

The presence or absence of orthologues for the 54 conoid-associated proteins found in 157 Alveolata taxa is displayed in Fig 6, with proteins clustered according to their phylogenetic distributions. This orthology inventory shows that *T. gondii* conoid-associated proteins are most highly conserved in other coccidians. In Sarcocystidae (other than *Toxoplasma*), average completeness is 92%, whereas in the Eimeriidae, it is 69% (S3 Table). In other major apicomplexan groups, the average representation of the *T. gondii* conoid-associated proteins are: *Plasmodium* spp. 53%, Piroplasmida 33%, *Cryptosporidium* spp. 50%, and Gregarinasina 36% (but over 60% for some taxa). It is noteworthy that *Cryptosporidium* spp. and gregarines possess conspicuous conoids and that they share a similar subset of the *T. gondii* conoid proteome with members of the Aconoidasida. Furthermore, these common proteins include proteins that locate to all specific conoid substructures in *T. gondii*: conoid base, body, and conoid canopy. Taxon-specific absences of *T. gondii* orthologues could represent protein loss in those taxa, gain of novel proteins specific to coccidians, or rapid evolution of the primary protein sequence that results in failure of orthologue detection. Collectively, however, these data support the conoid

and associated structures as being conserved throughout apicomplexans, including members of the Aconoidasida.

Many putative orthologues of conoid-related proteins are also found in the related clades of Myzozoa (Fig 6, S3 Table). Apicomonada, which includes the nearest photosynthetic relatives of apicomplexans such as *Chromera velia*, have on average 33% of the *T. gondii* proteins and up to 53% for the free-living predatory *Colpodella angusta*. Dinoflagellates have on average 48% of *T. gondii* proteins but up to 70% for some early-branching parasitic taxa (*Amoebophyra* spp.). Molecular evidence in many of these taxa is based on RNA-Seq so might be less complete than when genomic data is available, as is suggested by lower BUSCO scores (this is also the case for many gregarines) (Fig 6, S3 Table). Nevertheless, there is strong evidence of conservation of the core conoid proteome in these clades also (Fig 6). These data are consistent with ultrastructural evidence of a conoid and apical complex involved in feeding and parasitism in these taxa [5,57]. Ciliates show evidence of fewer of the conoid proteins being present, yet some are found even in this basal clade of Alveolata. There is further evidence of broadly conserved apical proteins in alveolates detected by our spatial proteomic approaches (95 proteins; S5 Fig, S3 Table), but many of these remain to have their specific apical locations determined.

## Conoid proteins locate to apical rings in *Plasmodium* zoites

To test if orthologues of *T. gondii* conoid-associated proteins occur in equivalent apical structures in *Plasmodium*, 9 orthologues were selected for reporter tagging in *P. berghei* (Table 1). This model provided ready access to all 3 invasive zoite forms of the parasite: the ookinete and sporozoite forms that occur in the mosquito vector, and the merozoite form of the mammalian blood-staged infection. The 9 proteins represented the 3 sites associated with the conoid (base, walls, and canopy) as well as APR1 and APR2 (PBANKA_907700 and PBANKA_1334800, respectively). Green fluorescent protein (GFP) fusions of these proteins were initially observed in the large ookinete form by live-cell widefield fluorescence imaging, and an apical location was seen for all (Fig 7A). Eight of these proteins were resolved only as a dot or short bar at the extreme apical end of the cell, whereas the APR2 orthologue (PBANKA_1334800) presented as an apical cap.

To further resolve the location of the *P. berghei* apical proteins, 3D-SIM was performed on fixed ookinetes for 8 proteins representing the different presentations found in *T. gondii*. The *P. berghei* orthologue of the conoid wall protein (PBANKA_0310700) was resolved as a ring at the cell apex, and this structural presentation was also seen for orthologues of the conoid base (PBANKA_1216300) and canopy rings (PBANKA_1347000, PBANKA_1419000) (Fig 7B). Further, 2 orthologues that are unresolved conoid canopy puncta in *T. gondii* are seen in *P. berghei* to mimic this presentation either as an apical punctum (PBANKA_1025300) or a barely resolved small ring (PBANKA_1313300) (Fig 7B). The APR2 orthologue (PBANKA_1334800) that showed a broader cap signal by widefield imaging was revealed as a ring of much larger diameter than the rings of the conoid orthologues (Fig 7B). Furthermore, short spines radiate from this ring in a posterior direction that account for the cap-like signal at lower resolution. The location of this protein is consistent with an APR2 function, although more elaborate in structure than what is seen in *T. gondii* (see Fig 5B). Finally, the APR1 orthologue (PBANKA_0907700) also resolved as a ring of larger diameter than the conoid orthologues and apparently closer to the apical cell surface than APR2 orthologue PBANKA_1334800 (Fig 7B). In all cases examined, the locations and structures formed by the *Plasmodium* orthologues were equivalent to those of *T. gondii*, strongly suggestive of conservation of function.

Transmission electron micrographs (TEMs) of *P. berghei* ookinetes further support the presence of conoidal ring structures implied by our proteomic data and fluorescence

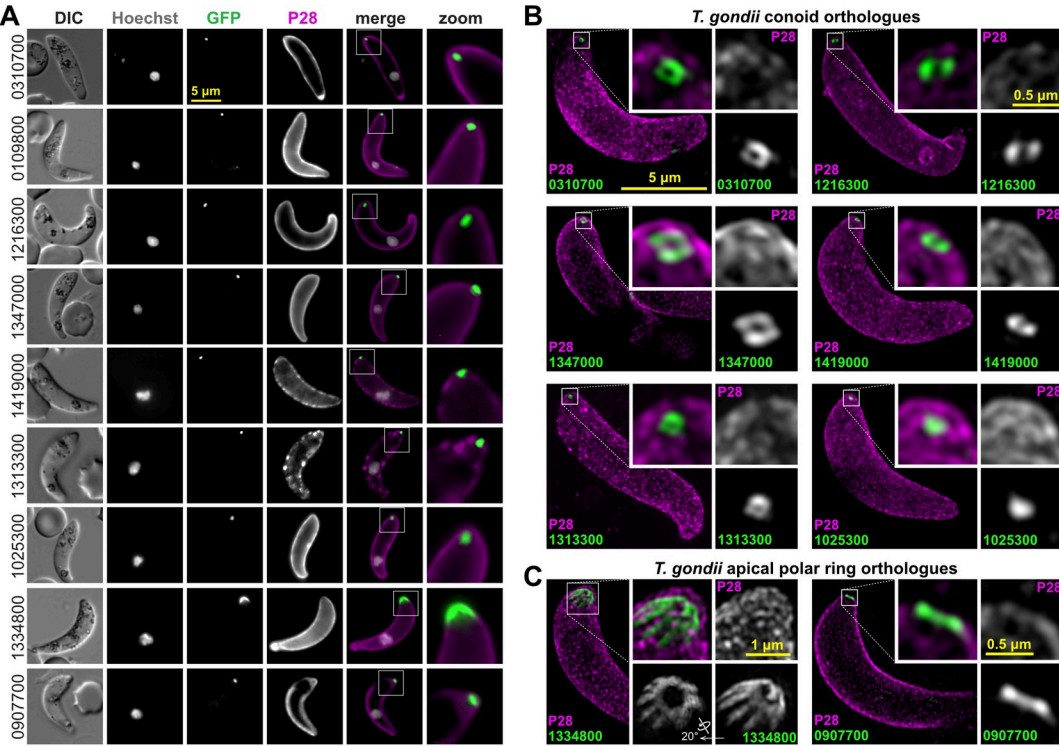

**Fig 7. Live-cell widefield and super-resolution imaging of *P. berghei* ookinetes expressing GFP fusions of conoid complex orthologues.** *T. gondii* orthologue locations are shown in Figs 3–5. (A) Widefield fluorescence imaging showing GFP (green), Hoechst 33342-stained DNA (grey), and live cy3-conjugated antibody staining of ookinete surface protein P28 (magenta). (B, C) 3D-SIM imaging of fixed GFP-tagged cell lines for conoid orthologues (B) or APR orthologues (C) with same colours as before (A). Inset for APR protein (1334800) shows rotation of the 3D-reconstruction to view the parasite apex face on. All panels are at the same scale, scale bar = 5 μm, with zoomed inset from yellow boxes (inset scale bar = 0.5 μm or 1 μm for 1334800). 3D-SIM, three-dimensional structured illumination microscopy; APR, apical polar ring; DIC, differential interference contrast; GFP, green fluorescent protein.

microscopy (Fig 8, S6 Fig). At the apex of *Plasmodium* ookinetes, the IMC and subpellicular microtubules are separated by a thick collar that presents as an outer electron-dense layer and an inner electron-lucent layer (Fig 8B–8F). This collar displaces the APR2 approximately 100 nm posterior to APR1. Within the APR1, 3 further rings can be seen in either cross section or tangential section, the posterior of the 3 rings is thicker than the anterior two (Fig 8B–8D). The most apical of these rings is often seen to distend the plasma membrane creating a small apical protrusion, and thin ducts from the micronemes can be seen extending through all 3 of these rings to the plasma membrane (Fig 8A inset, 8B–8E). This ultrastructure is consistent with equivalent conoidal ring structures observed in *Toxoplasma* (Fig 1): 2 conoid canopy rings atop conoid walls that are reduced in height and skeletal components compared to *Toxoplasma* and other apicomplexans.

## Conoid-type structures are present but compositionally distinct between vector and mammalian *Plasmodium* zoite forms

The presence of a possible conoid in *Plasmodium* has been previously attributed to the ookinete-stage [26], but the conoid is widely considered to be absent from asexual blood-stage merozoites. With our new markers for components of apparent conoid-associated structures in *P. berghei*, we tested for presence and location of these proteins in the other zoite stages:

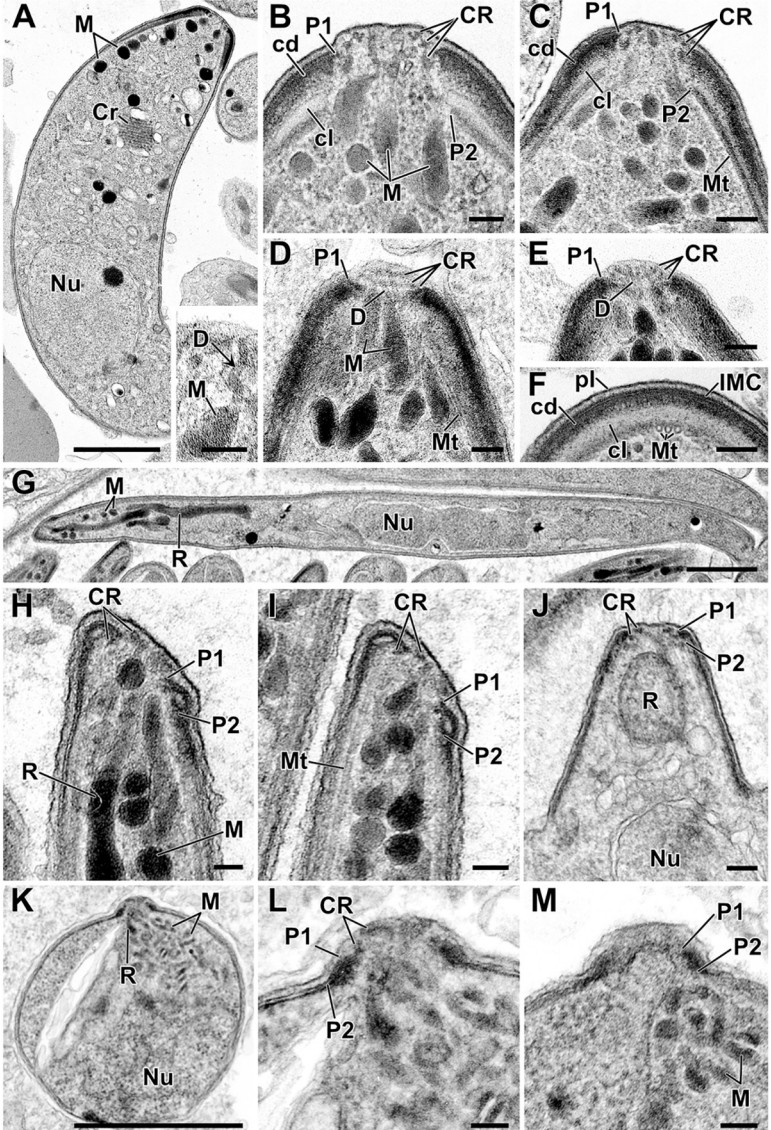

**Fig 8. Ultrastructure of conoid complexes of *P. berghei* zoites.** Transmission electron micrographs of *P. berghei* zoites: ookinetes (A–F), sporozoites (G–J), and bloodstream merozoites (K–M). (A) Longitudinal section through an ookinete showing the apical complex with micronemes (M) plus the crystalline body (Cr). Insert: Detail of the apical cytoplasm showing a microneme (M) with a duct running towards the anterior (arrows). (B–E) Details of longitudinal and tangential sections through the apical complex with either 2 or 3 CRs evident with the anterior collar consisting of an outer electron-dense layer (cd) closely adhering to the IMC which forms the anterior polar ring (P1) and an inner electron-lucent layer (cl) which is closely associated with subpellicular microtubules (Mt) which forms the inner polar ring (P2). Underlying micronemes (M) with ducts (D) extend to the cell apex. F. Cross section through part of the apical collar showing the ookinete plasma membrane (pl) with the underlying IMC closely adhering to the electron-dense layer of the collar (cd) with the more electron-lucent region (cl) closely associated with subpellicular microtubules (Mt). (G) Longitudinal section through a sporozoite showing the anteriorly located rhoptries (R) and micronemes (M) and the central nucleus (Nu). (H, I). Detail of the anterior of the mature sporozoites showing the CRs and the in-folding of the IMC to form the first APR (P1) with second APR beneath (P2) associated with the subpellicular microtubules (Mt). Note the angle formed by the apical polar rings relative to the longitudinal cell axis. (J) Longitudinal section of an early stage in sporozoite formation showing apical CRs and the perpendicular projection of the CRs and APRs. (K) Longitudinal section through a spherical-shaped merozoite released from an erythrocyte showing the rhoptries (R), micronemes (M), and nucleus (Nu). (L, M) Enlargement of the apical region showing the CRs and the closely positioned polar rings (P1, P2). Scale bars represent 1 μm (A, G, K) and 100 nm in all others. See also S6 and S7 Figs. APR, apical polar ring; CR, conoidal ring; IMC, inner membrane complex.

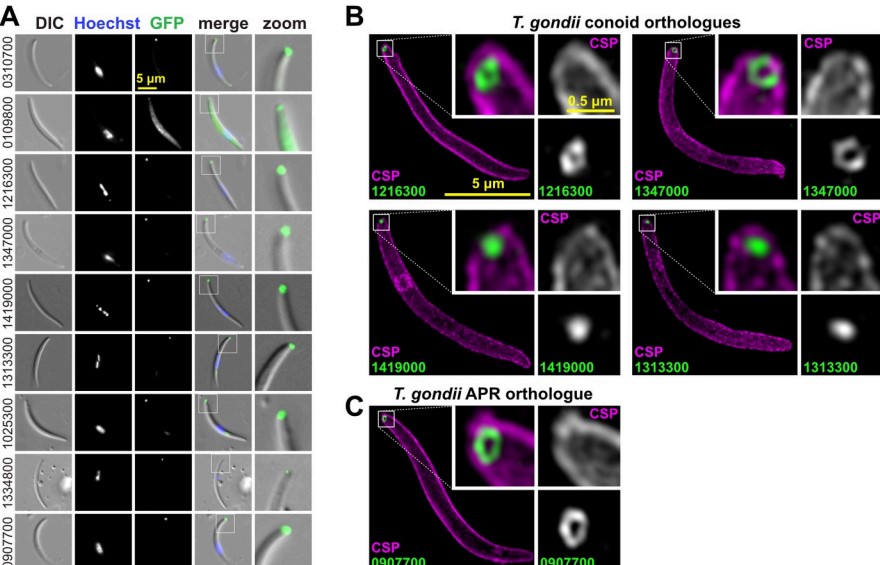

**Fig 9. Live-cell widefield and super-resolution imaging of *P. berghei* sporozoites expressing GFP fusions of conoid complex orthologues.** (A) Widefield fluorescence imaging showing GFP (green) and Hoechst 33342-stained DNA (grey). All panels are at the same scale, scale bar = 5 μm, with the exception of zoomed images from white boxed regions in the merge. (B, C) Super-resolution imaging of GFP-fused conoid complex proteins (green) in fixed cells shown with the cell surface stained for sporozoite surface protein CSP (magenta). All panels are at the same scale, scale bar = 5 μm, with zoomed inset from white boxed regions (inset scale bar = 0.5 μm). CSP, circumsporozoite protein; DIC, differential interference contrast; GFP, green fluorescent protein.

sporozoites and merozoites (Fig 8G–8M). In sporozoites, all proteins tested for are detected at the cell apex (Fig 9A), and super-resolution imaging of 5 of these again showed either a ring or unresolved apical punctum (Fig 9B).

In merozoites, of the 9 proteins tested for, only 6 were detected in this alternative zoite form of the parasite, and this is generally consistent with differential transcript expression profiles of these 9 genes (Table 1, S8 Fig). The conoid wall (PBANKA_0310700) and base (PBANKA_1216300) orthologues were not detected in this cell form, nor was the APR2 protein (PBANKA_1334800). However, all 5 of the other conoid orthologues are present in merozoites as well as the APR1 protein (PBANKA_0907700), each forming an apical punctum juxtaposed to the nucleus consistent with apical location (Fig 10). These data support conservation of conoid constituents in the apical complex of both sporozoites and merozoites, but either a reduction in the complexity of this structure in merozoites or the possible substitution for other proteins that are yet to be identified.

## Discussion

The discovery of the conoid was one of the early triumphs of electron microscopy applied to thin biological samples. The term "conoid" was coined by Gustafson and colleagues in 1954 to describe the hollow truncated cone observed first in *Toxoplasma* [58]. They described this structure as having "no close anatomical parallel... in other protists", and it provided the first identification of the penetration device used by apicomplexan parasites. While the spiralling tubulin-rich fibres of the conoid wall attract most attention, this cell feature is actually part of a continuum of structures better described as the "conoid complex" (Fig 11) [5]. This complex starts at the apical limits of the IMC coordinated by the 2 APRs [23,59,60]. The conoid is tethered by its base within these APRs [24] and is in close proximity and probable interaction with

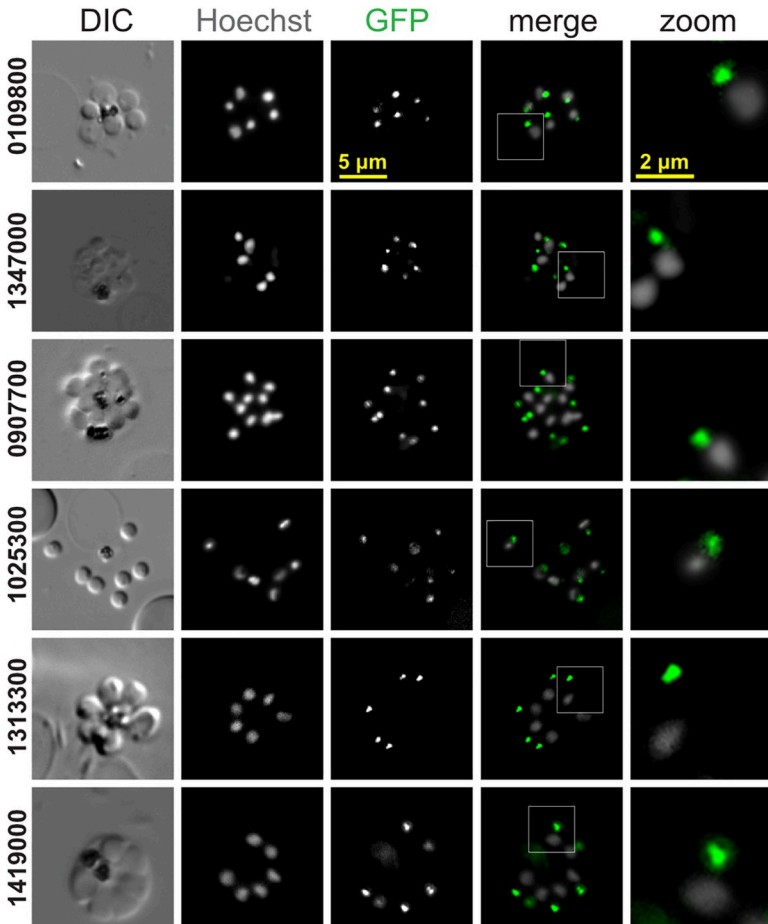

**Fig 10. Live-cell imaging of *P. berghei* merozoites expressing GFP fusions of conoid complex orthologues.**
Widefield fluorescence imaging showing GFP (green) and Hoechst 33342-stained DNA (grey) with some parasites seen pre-egress from the erythrocyte and others post egress. All panels are at the same scale, scale bar = 5 μm shown, with zoomed inset from white boxed regions (inset scale bar = 2 μm). DIC, differential interference contrast; GFP, green fluorescent protein.

the plasma membrane via the apical conoid canopy. Proteins have been previously located to all of these "substructures" including some linking one substructure to the next [5]. Indeed, the apparent spatial separation of compartments by ultrastructure is smaller than the size of many of the individual molecules that build it [24]. Thus, at a molecular level, it is unclear what the limits of any one substructure are, if this is even a biologically relevant notion.

In this study, we have provided a substantial expansion of knowledge of the molecular composition and architecture of the conoid complex in *T. gondii*. Previous identification of conoid complex proteins used methods including subcellular enrichment, correlation of mRNA expression, and proximity tagging (BioID) [30,31,44]. Among these datasets, many components have been identified, although often with a high false-positive rate. For example, the seminal conoid proteomic work of Hu and colleagues (2006) [44] detected approximately half of the proteins that we report (49 of 95, see S1 Table). However, in their study, a further 329 proteins that fractionated with the conoid (≥2-fold enriched; ToxoDB Release 49) included many identifiable contaminants including known cytosolic, ribosomal, mitochondrial, apicoplast, and microneme proteins. We have found the hyperLOPIT strategy to be a powerful approach for enriching for proteins specific to the apex of the cell, and BioID has further

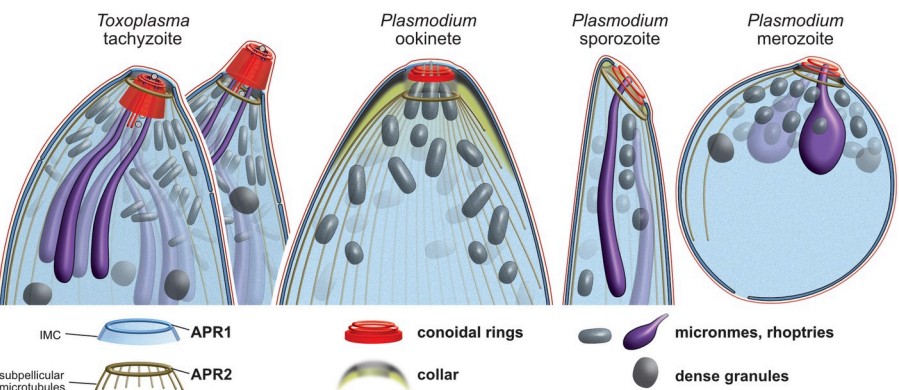

**Fig 11. Conservation and variability of the conoid complex in apicomplexan zoite forms.** Schematics of cell apices from *Toxoplasma* and *Plasmodium* showing presence of common structures but displaying variability in their size and arrangement. *Toxoplasma* is shown with either the conoid retracted or protruded. A row of vesicles of unknown function lines the intraconoidal microtubules in *Toxoplasma* and other coccidians. Schematics draw on both TEM and EM tomography data that is presented or cited throughout the report. APR1, apical polar ring 1; APR2, apical polar ring 2; EM, electron microscopy; IMC, inner membrane complex; TEM, transmission electron micrograph.

refined identification of proteins specific to the conoid complex region. Collectively, we now know of 54 proteins that locate at the conoid complex (Table 1), with a dataset of many further proteins that await closer inspection (S1 Table). Moreover, we have used high-resolution microscopy to define the specific locations of many, and these show dedicated locations from the APRs through to proteins tethered to the plasma membrane. These data reveal a molecularly complex feature well beyond its tubulin component.

The conservation of a large part of the conoid complex proteome throughout Apicomplexa suggests that this is a cell feature maintained throughout the phylum. Conservation includes proteins from all substructural elements suggesting the maintenance of this structure in its entirety rather than only select elements. Where clade-specific losses of sets of genes are seen, these are not enriched for specific conoid complex locations that would indicate losses of select substructures (Fig 6, S5 Fig). It is to be noted that, in some cases, the predicted absence of proteins might represent false negative due to extreme protein divergence. While our reverse BLASTp criterion performed well to prevent inclusion of nonhomologous proteins and distant paralogues, it might have eliminated highly divergent, but bona fide, orthologues. The phylogenetic distributions of identifiable conoid complex proteins may also provide some clues to protein function. Proteins that interact or contribute to a common molecular function are likely to coevolve. The phylogenetic distributions of presence, absence, or divergence of conoid complex proteins might, therefore, provide evidence of molecular cooperation, including that spanning or linking conoid complex substructures. Gene knockout studies in both *T. gondii* and *P. falciparum* indicate that proteins in all parts of the conoid complex play key roles in parasite viability, including in the blood-stage of *Plasmodium* that has an apparently reduced or modified conoid complex (Table 1). Collectively, our data strongly suggest the conservation of the conoid complex in Aconoidasida, including in the piroplasms where microscopy has, so far, also failed to identify some of these structures.

Conoid complex proteins are also seen in Apicomplexa's sister clades, most notably in other lineages of Myzozoa. These data provide the first molecular support for previous hypotheses that the similar ultrastructure of "apical complexes," that function in both parasitism and predation in these myzozoan relatives, represent genuine homologous structures of the apicomplexan conoid complex [5,12]. The presence of some conoid complex protein homologues

in ciliates and the ancestral free-living predatory colponemids might indicate an even more ancient origin of this structure, or perhaps the repurposing of existing ancestral proteins into a derived apical complex structure in apicomplexans. Further protein phylogenetic analyses combined with proteomics and microscopy in non-apicomplexan lineages will be necessary to test these hypotheses.

The prior interpretation of a conoid being absent in *Plasmodium* stages, and in other Aconoidasida, mostly stems from the lack of a conspicuous extended tubulin fibrous conoid wall as seen by electron microscopy in coccidians, *Cryptosporidium* spp. and gregarines. A microtubular component of the conoid, however, has been reported from other members of order Haemosporida such as in ookinetes of *Leucocytozoon* and *Haemoproteus*, although in dramatically reduced form [59,61]. In both taxa, 3 conoidal rings are present with the posterior one containing microtubules. In *Leucocytozoon*, only a few microtubules remain, observable in longitudinal section but with some difficulty due to the surrounding density of other molecules. With any further reduction in the tubulin component in *Plasmodium* spp., or other Aconoidasida, detection of conoid tubulin by ultrastructural approaches would be even more challenged. However, the very recent application of ultrastructure expansion microscopy (U-ExM), in combination with anti-tubulin staining, has revealed that a ring of tubulin is present in *P. berghei* ookinetes at the very apex of the cell, beyond the apical termini of the subpellicular microtubules at APR2 [62]. This position is consistent with the location of the 3 conoidal rings we observe by TEM (Fig 8B–8D), and the thicker posterior conoidal ring is the most likely location of this tubulin given its presence here in *Haemoproteus* and *Leucocytozoon*. It is unknown if this tubulin forms a canonical microtubule, or a modified fibre such as that in the *Toxoplasma* conoid. Nevertheless, it is now apparent that there can be tubulin components of the apical complex ring(s) in apicomplexans that have previously evaded detection by electron microscopy. This presence of tubulin in a *Plasmodium* conoid complex provides additional support to our data showing the conservation of numerous conoid-associated proteins in all *Plasmodium* zoite forms. We have previously shown that SAS6L and Myosin B locate to an apical ring in *Plasmodium* also [42,63], and these colocate with the tubulin ring in *Plasmodium* ookinetes [62].

Collectively, these data suggest that the core architectural and compositional elements of the conoid complex are present in most apicomplexan zoites, although with variation in the size and presentation of some of these features (Fig 11). The apical polar rings APR1 and APR2 manage the apical opening in the cell pellicle. In *Plasmodium* ookinetes, the conspicuous thick collar defines the separation of the IMC from the subpellicular microtubules. We note that in *Plasmodium* sporozoites, an annular plaque of electron-lucent material is also present and corresponds to a tight reorientation of the apical IMC with respect to the APR2 (Fig 8H and 8I, Fig 11, S7 Fig) [64]. In merozoites, and also *Toxoplasma* tachyzoites, APR1 and APR2 are even closer together than in these other zoites; however, it is likely that some proteinaceous network manages their relative positions also. We note electron density between these rings in both of these zoite forms that might provide this function (Fig 1B and 1C, Fig 8K–8M, S7 Fig). In support of a common "collar," the APR protein (PBANKA_1334800) apparently contributes to the collar of ookinetes as spines (Fig 7) reminiscent of the translucent columns, so-called "tines," of ookinetes of other Haemosporida species [59,60,65]. In sporozoites and *T. gondii* tachyzoites, this protein also forms a ring, although without the spines, consistent with the contraction of this collar structure. Our study has identified multiple other *T. gondii* APR proteins with distinct anterior or posterior positions at the sites of these rings. Caution is required making inferences of precise protein occupancy with protein terminal reporters and the high spatial resolution achieved by 3D-SIM [24]. Nevertheless, our data suggest that some proteins might be specific to APR1, some to APR2, and some are at intermediate positions

that might represent further collar components. These subtle differences in APR protein location seen in *T. gondii* are consistent with the positions of their orthologues in *P. berghei* (Fig 7B, PBANKA_1334800 versus PBANKA_090770).

Within the APRs of all *Plasmodium* zoite forms are further conoidal rings, variously named "apical rings," "apical polar rings," or "polar rings" in past literature for previous lack of recognisable identity [25,27,66]. Three discernible conoidal rings in ookinetes are consistent with substantial contraction of the tubulin-containing conoid body walls and persistence of the 2 apical conoidal rings (Fig 11). In sporozoites, the number of conoidal rings is less clear; we discern at least two (Fig 8H–8J), although others have suggested more [67,68]. In merozoites, there are more clearly only two [27,69]. It is unknown if this reduction represents loss or merger of conoidal rings. However, the presence in all of these *Plasmodium* forms of proteins that occur at the base, walls, and canopy of the *Toxoplasma* conoid suggests compression of the overall conoid rather than loss of distinct elements.

The variation in the length of the conoid in *Plasmodium* compared with that seen in coccidians, *Cryptosporidium* and gregarines might reflect different mechanical properties and needs for these cells within their target host environments. It is presumed that the conoid in *Toxoplasma*, with its pronounced motility, provides a mechanical role in invasion. It is unknown if this high level of motility is seen more widely in other apicomplexans, but it might be an adaptation in *Toxoplasma* to the tremendously wide range of hosts and cell types that its zoites must invade, including penetrating thick mucilaginous layers at the gut epithelium. A reduction in this structural and mechanical element of the *Plasmodium* conoid complex likely reflects either the different invasion challenges presented by its hosts or different solutions that these parasites have evolved.

Evolutionary change in the architecture and composition of the conoid complex across taxa is further supported by differentiation of its proteome between the different *Plasmodium* zoite forms. We observe the blood-stage merozoite conoid proteome to be further reduced, or modified, when compared to ookinetes and sporozoites, and we previously observed SAS6L to be also absent from merozoites but present in ookinetes and sporozoites (Table 1) [42]. The differentiation of the merozoite apical complex also includes the absence of the APR2 protein that extends into the collar. Perhaps this elaboration of the collar in ookinetes is a *Plasmodium* adaptation in lieu of the extendibility of the conoid complex displayed by *Toxoplasma* and other coccidians [17]. In *Plasmodium*, these compositional and structural differences have likely been produced by the different zoite stages' invasion requirements between merozoites entering erythrocytes and the multiple penetration events for the ookinetes to reach the mosquito gut basal lamina, or the sporozoites to reach this vector's salivary glands and then mammalian hepatocytes [70]. Ookinete and sporozoite apical complexes might be under further selection by their need for high invasion competence. Each stage represents a transmission bottleneck with success among only one or few parasites required for onward transmission [71]. Increased investment in a robust and reliable apparatus might be justified at these important transmission points.

Evidence of conserved elements of the conoid and conoid complex throughout Apicomplexa, despite differences in construction and ultrastructure, raises the question of what are the functions of this structure and are they common throughout the phylum? Indeed, even in *Toxoplasma* the function of the conoid is relatively poorly understood. Most studies of individual protein elements have identified molecules required for its assembly and stability [29,31–35]. But other studies have implicated roles in control processes, including activating motility and exocytosis, both of which are requirements for invasion as well as host egress events [24,28,30]. Indeed, the conoid is intimately associated with both exocytic vesicles and the apex of the plasma membrane, and this is a common trait throughout not just Apicomplexa but

other myzozoans including perkinsids and dinoflagellates [18–20,72]. The conoid complex proteome is enriched for domains that mediate protein–protein interactions as well as responses to regulatory molecules (e.g., $Ca^{2+}$, cyclic nucleotides) or regulatory protein modifications, and these features are seen in many of the proteins conserved widely among taxa (Table 1, Fig 6). This speaks to the conoid complex comprising an ordered regulatory platform for control of vesicle trafficking, fusion and fission, as well as initiation of cell motility. Such a feature as this seems unlikely to be superfluous and lost in these parasites so heavily dependent on mediating complex interactions with hosts through this portal in an otherwise obstructed elaborate cell pellicle. Recognising the common components and importance of the conoid complex throughout Apicomplexa is highly relevant to understanding the mechanisms of invasion and host interaction and the pursuit of better drugs and intervention strategies to combat the many diseases that they cause.

## Methods

### Growth and generation of transgenic *T. gondii*

*T. gondii* tachyzoites from the strain RH and derived strains, including RH Δku80/TATi [82], were maintained at 37˚C with 10% $CO_2$ growing in human foreskin fibroblasts (HFFs) cultured in Dulbecco's Modified Eagle Medium supplemented with 1% heat-inactivated fetal bovine serum, 10 unit $ml^{-1}$ penicillin, and 10 μg $ml^{-1}$ streptomycin, as described elsewhere [83]. When appropriate for selection, chloramphenicol was used at 20 μM and pyrimethamine at 1 μM. Scaling up of the parasite culture for hyperLOPIT experiments was done according to the method described by [83]. Reporter protein-tagging of endogenous gene loci was done according our previous work [36] with oligonucleotides shown in S2 Table.

### BioID

**Sample preparation.** For the proximity biotinylation assay, we generated 3 different cell lines (*T. gondii* tachyzoites RH Δku80 TATi) by in situ genomic C-terminal tagging of one of the 3 bait proteins (SAS6L, RNG2, or MORN3) with the promiscuous bacterial biotin ligase, BirA*. The protein BirA*-tagging method used is described in our previous work [42] with oligonucleotides shown in S2 Table. We then followed the BioID protocol of Chen and colleagues [73]. The biotinylation of the proximal proteins by the BirA* enzyme was promoted by addition of 150 μM biotin into the ED1 growth media 24 h prior to parasite egress. The nontagged parental cell line was used as a negative control for background biotinylation. The BirA* activity was validated by a western blot detection of the biotinylated proteins. Samples of $1 \times 10^7$ tachyzoites were lysed in the NuPAGE LDS Sample Buffer (ThermoFisher Scientific, Waltham, Massachusetts, United States of America) and separated in the NuPAGE 4–12% Bis-Tris gel (ThermoFisher) and blotted on a nitrocellulose membrane, which was then blocked by 3% BSA for 1 h. The membrane was then incubated in the presence of streptavidin-HRP conjugate (ThermoFisher; 1:1000 dilution in 1% BSA) for 1 h, followed by five 5-min washes in TBST (tris-buffered saline solution containing 0.05% (w/v) of Tween 20). The HRP chemiluminescent signal was visualised by the Pierce West Pico kit (ThermoFisher) and a digital camera.

For the proteomic analysis, approximately $2 \times 10^9$ tachyzoites were harvested after 24-h biotinylation and egress. The parasites were separated from the host-cell debris by passing through a 3-μm filter and washed 5× in phosphate-buffered saline. The cell pellets were lysed in RIPA buffer, and the volume of each sample was adjusted to 1.8 mL and 5 mg of total protein content. A volume of 200 μL of unsettled Pierce Streptavidin Magnetic Beads (Thermo-Fisher) were first washed in RIPA buffer and then incubated with the cell lysates for 1 h at room temperature with gentle rotation of the tubes. The beads were then washed 3× in RIPA,

1× in 2M UREA 10 mM Tris-HCl (pH 8.0), and again 3× in RIPA, reduced in 10 mM DTT and 100 mM ammonium bicarbonate solution for 30 min at 56˚C, alkylated in 55 mM iodoacetamide and 100 mM ammonium bicarbonate for 45 min at room temperature in the dark, and finally washed in 50 mM ammonium bicarbonate for 15 min at room temperature with gentle rotation. The peptides were digested off the beads by an overnight 37˚C incubation in 1 μg of trypsin dissolved in 50 μl of 50 mM ammonium bicarbonate, and the remaining peptides were extracted with 100 μl of 10% formic acid. Both peptide extractions were combined, desalted using C18 solid-phase extraction cartridges (SepPak C18, 100 mg sorbent, Waters, Milford, Massachusetts, USA), dried in a vacuum centrifuge (Labconco, Kansas City, Missouri, USA), and resuspended in 0.1% (vol.) aqueous formic acid.

**Peptide analysis by liquid chromatography and mass spectrometry.** Peptide fractions were analysed in a Q Exactive hybrid quadrupole-Orbitrap mass spectrometer (Thermo Fisher Scientific, Waltham, Massachusetts, United States of America) coupled on-line with a nano-flow ultrahigh-performance liquid chromatography (nano-UPLC) system Dionex UltiMate 3000 RSLCnano (Thermo Fisher Scientific, Waltham, Massachusetts, USA) through an EASY-Spray nano-electrospray ionization (nano-ESI) source (Thermo Fisher Scientific, Waltham, Massachusetts, USA). Samples were loaded onto an Acclaim PepMap 100 C18 trapping column (300 μm i.d. × 5 mm length, 5 μm particle size, 100 Å pore size; Thermo Fisher Scientific) from the UltiMate 3000 autosampler with 0.1% aqueous formic acid for 3 min at a flow rate of 10 μL/min. After this period, the column valve was switched to allow elution of peptides from the precolumn onto the analytical column. The trapped peptides were then separated on an analytical reverse-phase nano EASY-Spray RSLC C18 column (75 μm i.d. × 50 cm length, 2 μm particle size, 100 Å pore size; Thermo Fisher Scientific) at a flow rate of 300 nL/min using a gradient elution program. Solvent A was 0.1% (vol.) formic acid in water (HPLC-grade, Fisher). Solvent B was 0.1% (vol.) formic acid in water-acetonitrile (HPLC-grade, Rathburn, Walkerburn, United Kingdom) blend (20%: 80%, vol.). The linear gradient employed was 2% to 40% B in 30 min. Further wash and equilibration steps gave a total run time of 60 min.

The peptides eluting from the nano-LC column were ionized by applying a positive voltage of 2.1 kV to the nano-ESI emitter electrode. The mass spectrometer was operated in the data-dependent acquisition mode (DDA). The precursor survey mass spectra (MS1) were acquired between 380 and 1500 *m/z* at a 70,000 resolution with an AGC target of 1e6 and a maximum C-trap fill time of 250 ms. The top 20 most intense precursor ions were selected in the quadrupole using an isolation window of 1.5 *m/z* and subjected to fragmentation in the HCD mode at a normalised collision energy of 25. A dynamic exclusion limit of 20 s was applied. The fragment ion spectra (MS2, or MS/MS) were acquired between 100 and 1,600 m/z at a 17,500 resolution with an AGC target of 5e4 and a maximum C-trap fill time of 120 ms. The raw data were managed in XCalibur v2.1 (Thermo Fisher Scientific).

**LC-MS/MS data processing and protein identification.** The raw LC-MS/MS data were interpreted, and peak lists were generated using Proteome Discoverer 1.4 (Thermo Fisher Scientific). The peak lists were exported as Mascot generic files and searched against the annotated protein sequences of *Toxoplasma gondii* strain ME49 (ToxoDB.org release 29, retrieved on 20 December 2016; 8,322 sequences) and common contaminant proteins (cRAP, 115 sequences) with Mascot v2.6.0 (Matrix Science, Boston, Massachusetts, USA) assuming digestion by trypsin with up to 2 missed cleavage sites. The precursor and fragment ion tolerances were set to 5 ppm and 0.1 Da, respectively. Carbamidomethylation of cysteine was set as a static modification. Oxidation of methionine and deamidation of asparagine and glutamine were allowed as variable modifications. Scaffold (version Scaffold_4.7.5, Proteome Software, Portland, Oregon) was used to validate MS/MS-based peptide and protein identifications.

Peptide identifications were accepted if they could be established at greater than 95.0% probability by the Peptide Prophet algorithm [84] with Scaffold delta-mass correction. Protein identifications were accepted if they could be established at greater than 95.0% probability and contained at least 2 identified peptides. Protein probabilities were assigned by the Protein Prophet algorithm [85]. Proteins that contained similar peptides and could not be differentiated based on MS/MS analysis alone were grouped to satisfy the principles of parsimony. Total spectrum counts were used to calculate the enrichment of proteins in the BioID pulldown samples relative to the wild-type control sample. The contaminant proteins were not considered for the analysis.

### hyperLOPIT spatial proteomics

The hyperLOPIT spatial proteomics data were acquired and analysed as described in [36]. Three independent hyperLOPIT datasets contained 4,189, 4,547, and 4,292 proteins identified at FDR < 5% based on at least 1 unique high-confidence peptide (FDR < 1%) and quantified across all 10 TMT quantification channels. These three 10plex datasets were considered individually. In addition, the data on proteins shared across pairs or all 3 individual datasets were concatenated affording three 20plex (4,021, 3,916, and 4,078 proteins for 1∩2, 1∩3, and 2∩3, respectively) and one 30plex (3,832 proteins), thus yielding 7 datasets in total. For each dataset, the normalised protein abundance distribution profiles were derived from the TMT reporter ion intensity values. A Bayesian machine-learning classification method based on t-augmented Gaussian mixture models with maximum a posteriori prediction (TAGM-MAP) [86] was applied to probabilistically attribute proteins to 26 subcellular niches defined by 718 marker proteins. The protein localisation probability was calculated as the probability of a protein belonging to the most likely subcellular class and not being an outlier. Assignments at a localisation probability above 99% were accepted.

### Immunofluorescence microscopy

*T. gondii*-infected HFF monolayers grown on glass coverslips were fixed with 2% formaldehyde at room temperature for 15 min, permeabilized with 0.1% TritonX-100 for 10 min, and blocked with 2% BSA for 1 h. The coverslips were then incubated with a primary antibody for 1 h, followed by 1 h incubation with a secondary antibody. Coverslips were mounted using ProLong Diamond Antifade Mountant with DAPI (ThermoFisher Scientific, Massachusetts, USA). Images were acquired using a Nikon Eclipse Ti widefield microscope with a Nikon objective lens (Plan APO, 100×/1.45 oil) and a Hamamatsu C11440, ORCA Flash 4.0 camera.

3D structured illumination microscopy (3D-SIM) was implemented on either a DeltaVision OMX V4 Blaze (GE Healthcare, Issaquah, California, USA) or an Elyra 7 (Zeiss, Oberkochen, Germany) with samples prepared as for widefield immunofluorescence assay (IFA) microscopy with the exception that High Precision coverslips (Marienfeld Superior, No1.5H with a thickness of 170 μm ± 5 μm) were used in cell culture, and Vectashield (Vector Laboratories, Burlingame, California, USA) was used as mounting reagent. For protruded conoid imaging of extracellular parasites, cells were treated for 45 s with 5 μm 5- benzyl-3-iso-propyl-1H-pyrazolo[4.3-d]pyrimindin-7(6H)-one (BIPPO) prior to fixation. Samples were excited using 405, 488, and 594 nm lasers and imaged with a 60x oil immersion lens (1.42 NA). The structured illumination images were reconstructed in softWoRx software version 6.1.3 (Applied Precision). All fluorescence images were processed using Image J software (http://rsbweb.nih.gov./ij/).

**Electron microscopy.**    Samples of extracellular *T. gondii* tachyzoites and *P. berghei* ookinetes, oocysts, sporozoites, and merozoites were fixed in 4% glutaraldehyde in 0.1 M

phosphate buffer and processed for electron microscopy as previously described [87]. Briefly, samples were postfixed in osmium tetroxide, treated en bloc with uranyl acetate, and dehydrated and embedded in Spurr's epoxy resin (TAAB Lab Supplies, UK). Thin sections were stained with uranyl acetate and lead citrate prior to examination in a JEOL1200EX or JEOL JEM1400 electron microscope (Jeol, UK).

## Detection of putative orthologues in other alveolates

**Exhaustive expansion of identified homologues.** To search in-depth for homologues across the Alveolata, an in-house set of 419 proteomes belonging to the Stramenopila–Alveolata–Rhizaria (SAR) clade was used. OrthoFinder [56] was applied to cluster these proteins into OrthoGroups (i.e., sets of proteins that are descended from a single protein in the last common ancestor of SAR). For each of the 107 putative apical proteins ("queries") of *Toxoplasma gondii* ME-49 (these 107 proteins correspond to the union of the proteins in Table 1 and S1 Table), the OrthoGroup to which it belonged was identified. Of note, 4 pairs of *T. gondii* query proteins belonged to the same OrthoGroup (see S3 Table), indicating that these pairs of apical localising proteins are likely paralogues of one another. Subsequently, the presence-absence patterns for each of these OrthoGroups across the SAR were assessed. Based on the initial OrthoFinder presence-absence patterns, the OrthoGroups were separated into 2 categories in order to apply distinct search strategies: (1) OrthoGroups with a limited distribution, not present in clades other than apicomplexans; and (2) OrthoGroups with a wide distribution, present across alveolates, often with many species having multiple homologues. The search strategy corresponding to homologues of each *T. gondii* query can be found in S3 Table.

For search strategy (1), the protein sequences were filtered using PREQUAL v.1.02 [88] to identify and mask regions with nonhomologous adjacent characters, aligned using MAFFT v7.471 (option "auto") [89], and a profile HMM was built using hmmbuild from HMMER 3.3 (http://hmmer.org). This custom profile HMM was used to search for divergent homologues in the SAR proteome database using hmmsearch (E-value cutoff for the best scoring domain: 0.01); these were added to the OrthoGroup. The expanded OrthoGroup was then submitted to additional rounds of PREQUAL-MAFFT-hmmbuild-hmmsearch, until either (1) no new homologue was detected, or (2) the total number of accumulated OrthoGroup sequences exceeded 10,000, or (3) 10 rounds had been performed.

For search strategy (2), a single round of PREQUAL-MAFFT-hmmbuild-hmmsearch steps was applied with a more stringent E-value cut-off for the selection of new homologues, which was customised for each OrthoGroup. For this, the E-values of the sequences already in the OrthoGroup were retrieved. The highest of these E-values was used as the inclusion cutoff for new hits, unless the latter was higher than 0.001 (in which case, the E-value cutoff was set to 0.001).

**Filtering homologues for putative orthologues.** To exclude putative paralogues and or inclusion of spurious sequences by the HMM iterations, each sequence from the expanded OrthoGroups was used as a query for a sequence similarity search against the *T. gondii* proteome using BLASTp from BLAST 2.10.1+ [90] (E-value <10). If a query yielded a different *T. gondii* sequence as best hit, it was removed as a putative paralogue. Sequences that retrieved the original *T. gondii* sequence as best hit were annotated with "rbh" (for "reciprocal best hit"), while sequences that did not hit any sequence in *T. gondii* were annotated with "none." The latter can happen specifically in fast-evolving protein families, for which the highly sensitive HMM search was able to detect the homologue, contrary to a BLASTp search only. It is to be noted that, in some cases, putative orthologues (sequences that are clearly homologous to orthologues in other species) are too divergent to be retrieved as the best hit in a reverse

BLASTp search against the *Toxoplasma* proteome. Such cases may well present themselves in our dataset, and caution is particularly warranted when only a few species of a given clade have a detected putative orthologue and others of the same clade seemingly do not. The numbers of putative orthologues in the Alveolata subset of our SAR proteome dataset were inferred from these OrthoGroup sets (S3 Table). For the OrthoGroups that contained 2 query proteins (see "Exhaustive expansion of identified homologues"), the collected sequences were split and selected based on which of the 2 *T. gondii* proteins represented their best BLASTp hit. The "none" homologous sequences were retained for both protein queries.

The resulting putative orthologous sequences for each protein query can be found in FASTA files (S1 Data). The putative orthologues for *P. berghei* ANKA and *P. falciparum* 3D7 are listed by their GenBank and UniProt identifiers and original assembly gene names (S4 Table). For each alveolate species in our dataset, the completeness of its predicted proteome was inferred using BUSCO 4.0.5 with automatic selection of the reference lineage using the—auto-lineage-euk option [91]. These proteome completeness scores facilitate the interpretation of putative absences in the examined Alveolata species, for low completeness may indicate that a putative orthologue is not necessarily absent but may also simply result from an incomplete dataset (S3 Table, S5 Fig).

**Visualisation of putative orthologues across Alveolata.** For Fig 6 and S5 Fig, the binary presence/absence patterns of our putative orthology search were used (S3 Table) to calculated Jaccard distances for all protein pairs across the Alveolata, and proteins were clustered according to these distances using UPGMA. To generate the heatmaps in Fig 6 and S5 Fig, "heatmap2" of the R package gplots (https://cran.r-project.org/web/packages/gplots/index.html) was applied onto a "ternary" matrix, which distinguished between a protein being absent, it being present as a reciprocal best hit ("rbh"), or it being present as a none hit ("none") (see "Filtering homologues for putative orthologues"). To indicate the phylogenetic relationships of the species, a tree dendrogram without meaningful branch lengths was assembled by reviewing recent literature for Apicomplexa [92–94], Dinoflagellata [95–99], Ciliophora [100–106], and Colponemidia [107].

## Ethics statement

The animal work performed in the UK passed an ethical review process and was approved by the United Kingdom Home Office. Work was carried out in accordance with the United Kingdom "Animals (Scientific Procedures) Act 1986" and in compliance with "European Directive 86/609/EEC" for the protection of animals used for experimental purposes. The permit numbers for the project licences are 40/3344 and PDD2D5182.

## Generation of transgenic *Plasmodium berghei* and genotype analyses

To observe the location of *P. berghei* proteins, the C-terminus of the gene was tagged with green fluorescent protein (GFP) sequence by single crossover homologous recombination. To generate the GFP-tag line, a region of these genes downstream of the ATG start codon was amplified, ligated to p277 vector, and transfected as described previously [60]. The p277 vector contains the human *dhfr* cassette, conveying resistance to pyrimethamine. A schematic representation of the endogenous gene locus, the constructs and the recombined gene locus can be found in S9 Fig. A list of primers used to amplify these genes can be found in S2 Table. For the parasites expressing a C-terminal GFP-tagged protein, diagnostic PCR was used with primer 1 (Int primer) and primer 2 (ol492) to confirm integration of the GFP targeting construct. The primer details can be found in S2 Table.

### *P. berghei* phenotype analyses

Blood containing approximately 50,000 parasites of the GFP-tagged lines was injected intraperitoneally (IP) into mice to initiate infections. Parasitaemia was monitored by microscopy on Giemsa stained thin smears. Four to 5 days postinfection, blood was collected with 10% to 15% parasitaemia and allowed to further develop in schizont and ookinete culture medium as described previously [108]. The schizonts/merozoites and ookinetes were examined after 24 h of culture with a Zeiss AxioImager M2 microscope (Carl Zeiss) fitted with an AxioCam ICc1 digital camera. To examine the location of these proteins in sporozoites, 30 to 50 *Anopheles stephensi* SD 500 mosquitoes were allowed to feed for 20 min on anaesthetized, infected mice whose asexual parasitaemia had reached 15% and were carrying comparable numbers of gametocytes as determined on Giemsa stained blood films. On day 21 post feeding, 20 mosquitoes were dissected, and their guts and salivary glands crushed separately in a loosely fitting homogenizer to release sporozoites, which were then used for imaging.

## Supporting information

**S1 Table. *Toxoplasma* candidate apical proteins identified by hyperLOPIT and BioID methods.** Footnotes: [a]Known localization defined as "apex" when low-resolution imaging only has identified a punctum at the apex of the cell. AA, apical annuli; APR, apical polar ring; CCR, conoid canopy ring; ICMT, intraconoidal microtubules; PM, plasma membrane: [b]LOPIT assignment strength: • (black), Protein assigned to one of the 2 *apical* clusters at >99% probability in (before "/") the analysis of concatenated 3 or 2 datasets where the protein was full quantified or (after "/") the analysis of individual hyperLOPIT datasets; • (grey), as for above except apical assignment probability is below the 99% threshold; • (yellow), assignment to a cluster other than *apical* in 1 noncatenated hyperLOPIT datasets. For BioID proteomic experiments, the detection of a protein by a given bait is indicated by •. For Hu's and colleagues' 2006 proteomic study, proteins enriched with conoid fractions by at least 2-fold (ToxoDB Release 49) are indicated by •: [c]Mutant phenotype fitness scores where more strongly negative scores indicate increasing detrimental competitive growth in in vitro culture conditions for *T. gondii* [40].* References for localization data: [31–34,36,39,46,49,62–64,66,68,69,71–80]; TS, this study.
(XLSX)

**S2 Table. Oligonucleotide primers used for *T. gondii* and *P. berghei* gene tagging.** Footnotes: [a]PCR plasmid templates P5 and P6 as described in Barylyuk and colleagues (2020) [36]: [b]Endogenous gene fusion method for epitope-tagging or *birA**-tagging as described in Barylyuk and colleagues (2020) or Wall and colleagues (2016), respectively.
(XLSX)

**S3 Table. Sources and numbers of detected putative orthologues of apical proteins (Table 1, S1 Table) across Alveolata.** This table contains information on the alveolate species included in this study, including their full species and strain name, the data used, completeness of the predicted proteome using BUSCO, the percentages of conoid-associated (Table 1), and all apical (Table 1, S1 Table) proteins for which a putative orthologue was found in this species, and the numbers of putative orthologues for each individual protein.
(XLSX)

**S4 Table. GenBank/UniProt/PlasmoDB identifiers for putative *Plasmodium falciparum* 3D7 and *Plasmodium berghei* ANKA orthologues.** For *P. falciparum*, we had selected strain "IT" for our 419 SAR proteomes set. As a result, our sets of orthologues originally contained

sequences from this particular strain. However, because strain "3D7" is the one more commonly studied, we mapped the *P. falciparum* IT sequences to *P. falciparum* 3D7 by searching for reciprocal best blast hits among their proteomes. Therefore, *P. falciparum* 3D7 identifiers in the FASTA files (S1 Data) also provide information on which *P. falciparum* IT sequence was linked by.
(XLSX)

**S1 Data. FASTA files containing putative Alveolata orthologues for each *T. gondii* apical protein (Table 1, S1 Table, Fig 6, S5 Fig, Methods).** Files are named according to the *T. gondii* apical protein query. The sequence headers comprise the protein identifiers in our in-house proteome database for the SAR clade (Methods), starting with an abbreviation for the species strain (see S3 Table). The headers furthermore contain either the descriptor "self," indicating that this sequence is the original *T. gondii* query, or "rbh," indicating that it has the original *T. gondii* query as its best *T. gondii* hit in a BLASTp search, or "none," indicating that it has no BLASTp hit in *T. gondii* (see Methods). For sequences of *P. falciparum* 3D7, the header also indicates by which *P. falciparum* IT sequence it was gathered and if it was gathered via reciprocal best BLAST hits ("rbh") or via unidirectional best BLAST ("bh"). The latter is only the case for the putative orthologue of *T. gondii*'s TGME49_219070 (see also S4 Table).
(GZ)

**S2 Data. Differential transcript expression of *P. berghei* proteins in different lifecycle stages.** Mean normalised log counts of transcripts for each gene in different stages of *Plasmodium berghei* from single-cell RNAseq data [81] available in PlasmoDB.
(XLSX)

**S1 Fig. Immunodetection of candidate conoid complex proteins, identified by hyperLOPIT and BioID approaches, in *T. gondii* intracellular parasites.** Widefield fluorescence imaging of HA-tagged candidates (green) coexpressing either APR marker RNG2 or conoid marker SAS6L (magenta). All images are at the same scale, scale bar = 5 μm. APR, apical polar ring; BioID, proximity-dependent biotin identification; HA, hemagglutinin; hyperLOPIT, hyperplexed Localisation of Organelle Proteins by Isotope Tagging; SAS6L, SAS6-like.
(TIFF)

**S2 Fig. Super-resolution imaging of *T. gondii* proteins at the conoid body.** Immunodetection of HA-tagged conoid proteins (green) in cells coexpressing either APR marker RNG2 or conoid marker SAS6L (magenta) imaged either with conoids retracted within the host cell, or with conoids protruded in extracellular parasites. This figure shows further examples of conoid body proteins to those shown in Fig 3A. All panels are at the same scale, scale bar = 5 μm, with zoomed inset from yellow boxes (scale bar = 0.5 μm inset). APR, apical polar ring; HA, hemagglutinin; SAS6L, SAS6-like.
(TIFF)

**S3 Fig. Super-resolution imaging of *T. gondii* proteins at the conoid base, conoid canopy rings, and conoid canopy punctum.** Immunodetection of HA-tagged conoid proteins (green) in cells coexpressing either APR marker RNG2 or conoid marker SAS6L (magenta) imaged either with conoids retracted within the host cell, or with conoids protruded in extracellular parasites. This figure shows further examples of conoid proteins to those shown in Figs 3B, 4A and 5A. All panels are at the same scale, scale bar = 5 μm, with zoomed inset from yellow boxes (scale bar = 0.5 μm inset). APR, apical polar ring; HA, hemagglutinin; SAS6L, SAS6-like.
(TIFF)

**S4 Fig. Super-resolution imaging of *T. gondii* proteins at the apical polar rings.** Immunodetection of HA-tagged proteins (green) in cells coexpressing either APR marker RNG2 or conoid marker SAS6L (magenta) imaged either with conoids retracted within the host cell, or with conoids protruded in extracellular parasites. This figure shows further examples of apical polar ring proteins to those shown in Fig 5B. All panels are at the same scale, scale bar = 5 μm, with zoomed inset from yellow boxes (scale bar = 0.5 μm inset). APR, apical polar ring; HA, hemagglutinin; SAS6L, SAS6-like.
(TIFF)

**S5 Fig. Heatmap indicating conservation of putative orthologues of apical proteins among Alveolata.** Presences (red, orange) and absences (white) of putative orthologs of apical proteins in 157 surveyed Alveolata species (selection based on Table 1 and S1 Table, numbers indicate identifiers of the predicted proteome of *Toxoplasma gondii* ME49). In case of a presence, the species either contains at least 1 homologous sequence that has the *T. gondii* protein as its best BLASTp match (red) or it has only homologous sequences that did not yield any hit in *T. gondii* but were only obtained via (iterative) sensitive homology searches using HMMer (see Methods). The proteins were clustered based on the binary (presence-absence) patterns across the Alveolata, not making a distinction between the "red" and "orange" cells. Cluster visualisation was based on manual curation of readily identifiable clusters. Known localization defined as "apex" when low-resolution imaging only has identified a punctum at the apex of the cell. The species tree (top) outlines the phylogenetic relationships of the examined species, based on current literature (see Methods). The clades (bottom) are coloured according to the following clades: Apicomplexa sensu stricto (yellow), related Apicomonada (light-yellow), Dinoflagellata (blue), Perkinsozoa (green), Ciliophora (red), and Colponemidia (brown). For each of the species, the source of the protein predictions is indicated: genome (DNA, green) or transcriptome (RNA, dark red), and the BUSCO score indicates the completeness of these predictions. For detailed information on numbers and sources for this figure, see S3 Table. AC, apical cap; IMC, inner membrane complex; SPMT, subpellicular microtubules.
(EPS)

**S6 Fig. Ultrastructure of conoid complexes of *P. berghei* ookinetes.** Transmission electron micrographs of *P. berghei* ookinetes taken from Fig 8 with conoid complex features annotated in image duplicate. Scalebar = 100 nm. APR1, apical polar ring 1; APR2, apical polar ring 2; IMC, inner membrane complex.
(TIFF)

**S7 Fig. Ultrastructure of conoid complexes of *P. berghei* sporozoites and merozoites.** Transmission electron micrographs of *P. berghei* **A.** sporozoites and **B.** merozoites taken from Fig 8 with conoid complex features annotated in image duplicate. Scalebar = 100 nm for all except righthand panels of **B.** where scalebar = 1 μm. APR1, apical polar ring 1; APR2, apical polar ring 2.
(TIFF)

**S8 Fig. Differential transcript expression of *P. berghei* proteins in different lifecycle stages.** Mean normalised log counts of transcripts for each gene in different stages of *Plasmodium berghei* from single-cell RNAseq data [81]. See S2 Data for numeric values taken from PLasmoDB.
(TIFF)

**S9 Fig. Strategy and validation for GFP coding sequence integration in *P. berghei*.** (A) Schematic representation of the *P. berghei* endogenous gene locus, the integration constructs, and

the recombined gene locus. (B) Validation of correct gene tagging by diagnostic PCR.
(TIFF)

**S1 Raw Images. Original gel and blot images.**
(PDF)

## Acknowledgments

We are grateful to Mike Deery who performed the LC-MS/MS analysis of peptide samples,
Julie Howard Murkin for data processing of the BioID LC-MS/MS (both at the Cambridge
Centre for Proteomics), Emilie Daniel for technical assistance with *P. berghei* cell line genera-
tion, Yi-Wei Chang and Maryse Lebrun for useful discussions, and Sarah Marsden for com-
ments on the manuscript. We are grateful for support and assistance with super-resolution
imaging at the Cambridge Advanced Imaging Centre and the Gurdon Institute, and thank
Nicola Lawrence for imaging support.

## Author Contributions

**Conceptualization:** Ludek Koreny, Rita Tewari, Ross F. Waller.

**Data curation:** Ludek Koreny, Mohammad Zeeshan, Konstantin Barylyuk, Eelco C. Tromer, Jolien J. E. van Hooff, David J. P. Ferguson.

**Formal analysis:** Ludek Koreny, Mohammad Zeeshan, Konstantin Barylyuk, Eelco C. Tromer, Jolien J. E. van Hooff, Laura Eme.

**Funding acquisition:** Laura Eme, Rita Tewari, Ross F. Waller.

**Investigation:** Sara Chelaghma.

**Methodology:** Ludek Koreny, Mohammad Zeeshan, Konstantin Barylyuk, Eelco C. Tromer, Jolien J. E. van Hooff, Declan Brady, Huiling Ke, Rita Tewari.

**Software:** Jolien J. E. van Hooff.

**Supervision:** Laura Eme, Rita Tewari, Ross F. Waller.

**Validation:** Sara Chelaghma.

**Visualization:** Eelco C. Tromer, Jolien J. E. van Hooff.

**Writing – original draft:** Ross F. Waller.

**Writing – review & editing:** Ludek Koreny, Mohammad Zeeshan, Konstantin Barylyuk, Eelco C. Tromer, Jolien J. E. van Hooff, David J. P. Ferguson, Laura Eme, Rita Tewari, Ross F. Waller.

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
