## [Editor Report · Decision Letter 0]

30 Oct 2020

Dear Dr. Waller, 

Thank you for submitting your manuscript entitled "Conservation of the Toxoplasma conoid proteome in Plasmodium reveals a cryptic conoid feature that differentiates between blood- and vector-stage zoites" for consideration as a Research Article by PLOS Biology.

Your manuscript has now been evaluated by the PLOS Biology editorial staff as well as by an academic editor with relevant expertise and I am writing to let you know that we would like to move forward with it. However, before we can send you a decision letter, we need you to complete your submission by providing the metadata that is required for full assessment. To this end, please login to Editorial Manager where you will find the paper in the 'Submissions Needing Revisions' folder on your homepage. Please click 'Revise Submission' from the Action Links and complete all additional questions in the submission questionnaire.

Please re-submit your manuscript within two working days, i.e. by Nov 01 2020 11:59PM.

Kind regards,

Paula 

---

Associate Editor

PLOS Biology

---

## [Editor Report · Decision Letter 1]

4 Nov 2020

Dear Dr. Waller,

Thank you very much for submitting your manuscript "Conservation of the Toxoplasma conoid proteome in Plasmodium reveals a cryptic conoid feature that differentiates between blood- and vector-stage zoites" for consideration as a Research Article at PLOS Biology. Your manuscript has been evaluated by the PLOS Biology editors, and an Academic Editor with relevant expertise.

We are pleased to offer you the opportunity to address the comments from the reviewers in a revised version that we anticipate should not take you very long. We will then assess your revised manuscript and your response to the reviewers' comments and we may consult the reviewers again.

In particular, both the academic and in-house editors consider that the high resolution picture on Plasmodium asexual blood stages requested by referee #4 is important. Although addressing this issue would undoubtedly strengthen the study, given the amount of work involved in generating all cell lines necessary for analysing each protein with the two markers for relative localization, as you say in your plan for revision, this would not be strictly necessary for publication if not feasible to include. We also consider that you should provide the high resolution microscopy of the sporozoite and ookinetes. We consider that the suggestions for knock outs from referee #3 are not necessary for publication. 

The Academic Editor has further comments that you can see at the bottom of the letter, please address all of them. In summary, you should make explicitly clear whether the hyperlopit datasets analysed is the same as the Barylyuk paper and add that reference, acknowledge which of the proteins were previously assigned to the conoid by Hu as requested by referee #3, and add a supplementary figure with the actual conoidal rings outlined. 

Editorially, we would recommend you to be more concise in the introduction and discussion to facility the reading to a broader readership, and address the issue raised by the Academic Editor about the emotive language. 

We expect to receive your revised manuscript within 1 month.

**IMPORTANT - SUBMITTING YOUR REVISION**

*Resubmission Checklist*

*Published Peer Review*

*PLOS Data Policy*

*Blot and Gel Data Policy*

Sincerely,

Paula

---

Associate Editor,

pjaureguionieva@plos.org,

PLOS Biology

Comments from the Academic Editor:

The MS is clearly written, the authors integrate BioID with their existing data to present a convincing case with the orthogonal datasets corroborating the apical co-localisation of conoid associated proteins in T. gondii. The authors then use hi res microscopy to show that specific proteins are associated with particular Tg conoid substructures. Having thus expanded knowledge of the Tg conoid the authors show that nine of these proteins have an apical localisation in Plasmodium berghei and a handful were analysed by hi res microscopy to show ring-like structures or punctate apical structures. Overall the authors present a convincing case that the conoid is conserved in Plasmodium ookinetes and sporozoites and that some of the proteins are expressed in the apical complex of merozoites though the lack of hi-res microscopy hinders inferences of a conoid structure. The authors will have largely addressed the reviewers’ comments after inclusion of the promised additional hi-res microscopy of the sporozoite and ookinetes. 

The Barylyuk full reference is missing, this paper is online at cell host microbe. It is unclear from the submitted MS that the hyperlopit datasets analysed in the current MS are the same as those analysed in the barylyuk paper. This needs to be made explicitly clear when the hyperlopit data is first mentioned in the results.

The authors respond to R4’s comment that the intro and discussion could be more concise by citing the other reviewers comments on the MS’s eloquence. This is disingenuous as R2 described the text as “eloquent” but “wordy” and so in fact explicitly agreed with R4’s critique that the MS could be more concise. However, overall, I thought the text was clear and didn’t need revision, although there are occasional typos so I would recommend the authors carefully proof-read the MS. The first para of the intro uses more emotive language than is common in scientific texts, eg marauding, which I would avoid because I feel it implies intent by the parasite, however this is a matter of stylistic preference.

I’m not convinced by the authors arguments as to why they shouldn’t specifically acknowledge which proteins were previously associated with the conoid by Ke Hu. Although the hyperlopit plus BioID is more likely correct than the Hu conoid enrichment, it is still only confirmed by imaging for a handful of proteins and the assignment of the Hu predicted conoid proteins to other fractions is presumably also largely based on hyperlopit predictions. In the absence of definitive proof that the Hu “false positive” proteins are indeed localised elsewhere in the cell the authors cannot conclude that the Hu assignment was wrong and their hyperlopit assignment was correct. They should acknowledge which of the proteins were previously assigned to the conoid by Hu as requested by R3. They could include a statement indicating that according to the hyperlopit analysis there was probably a high background of false positives in the Hu conoid enrichment data with most conoid predicted proteins assigned to other cellular compartments. 

In figure 7 I found P1, P2 and the conoidal rings difficult to see, could the authors include a suppl fig with the actual structures outlined rather than just indicated by a bar?

Page 6

“The eight proteins represented the three sites associated with the conoid (base, walls and canopy rings) and one APR protein. GFP fusions of these proteins were initially observed in the large ookinete form by live cell widefield fluorescence imaging, and an apical location was seen for all (Fig 5)”

provide the accession in the text or label the figure or legend to indicate which is the apr protein

page 7

“Live reporter-tagging of this protein also reveals an apical punctum (Fig 5). The locations of all nine of these proteins are consistent with equivalent functions in the apical complex to those of T. gondii.”

label fig 5 or legend to identify this protein or provide accession here

“we tested for presence of these proteins in the other zoite stages: sporozoites and merozoites (Fig 7G-M)”

Is the wrong fig cited? fig 7 is TEM but not immuno-EM so no specific proteins were stained

“In sporozoites all proteins tested for are detected at the cell apex (Fig 8A)”

Provide panel of GFP merged with DIC to clearly show where the gfp colocalises with the spz apical structure

---

## [Editor Report · Decision Letter 2]

14 Dec 2020

Dear Dr Waller,

Thank you for submitting your revised Research Article entitled "Conservation of the Toxoplasma conoid complex proteome reveals a cryptic conoid in Plasmodium that differentiates between blood- and vector-stage zoites" for publication in PLOS Biology. I have now obtained advice from the Academic Editor. 

Given your revision, we will probably accept this manuscript for publication, assuming that you will modify the manuscript to address the data and other policy-related requests noted at the end of this email.

We suggest that you change the title of the manuscript to make it more accessible to non-experts: "In-depth characterization of the Toxoplasma conoid complex reveals its conservation in all apicomplexans, including Plasmodium species".

We expect to receive your revised manuscript within two weeks.

-  a cover letter that should detail your responses to any editorial requests, if applicable

*Published Peer Review History*

*Early Version*

Sincerely,

Paula

---

Associate Editor,

pjaureguionieva@plos.org,

PLOS Biology

DATA POLICY:

2) Deposition in a publicly available repository. Please make sure that the data is available by the time of publication.

Figure 6 and Supplementary figure 8.

**Please also ensure that figure legends in your manuscript include information on where the underlying data can be found, and ensure your supplemental data file/s has a legend.**

---

## [Editor Report · Decision Letter 3]

17 Dec 2020

Dear Dr. Waller,

On behalf of my colleagues and the Academic Editor, Michael Duffy, I am pleased to say that we can in principle offer to publish your Research Article "Molecular characterization of the conoid complex in Toxoplasma reveals its conservation in all apicomplexans, including Plasmodium species" in PLOS Biology, provided you address any remaining formatting and reporting issues.

Before your manuscript can be formally accepted, you will need to complete some formatting- and/or reporting-related requests that will be detailed in an email that will follow this letter. Please note that it usually takes a 2-3 business days for you to receive this email; during this time no action is required by you. Please note that your manuscript will not be formally accepted and scheduled for publication until you have made the required changes.

In the meantime, please log into Editorial Manager at http://www.editorialmanager.com/pbiology/, click the "Update My Information" link at the top of the page, and update your user information to ensure an efficient production process.

PRESS

Thank you again for supporting Open Access publishing. We look forward to publishing your paper in PLOS Biology. 

Sincerely, 

Paula

---

Paula Jauregui, PhD 

Associate Editor 

PLOS Biology